# Macropinocytosis confers resistance to therapies targeting cancer anabolism

Vaishali Jayashankar[1] & Aimee L. Edinger[1]✉

Macropinocytic cancer cells scavenge amino acids from extracellular proteins. Here, we show that consuming necrotic cell debris via macropinocytosis (necrocytosis) offers additional anabolic benefits. A click chemistry-based flux assay reveals that necrocytosis provides not only amino acids, but sugars, fatty acids and nucleotides for biosynthesis, conferring resistance to therapies targeting anabolic pathways. Indeed, necrotic cell debris allow macropinocytic breast and prostate cancer cells to proliferate, despite fatty acid synthase inhibition. Standard therapies such as gemcitabine, 5-fluorouracil (5-FU), doxorubicin and gamma-irradiation directly or indirectly target nucleotide biosynthesis, creating stress that is relieved by scavenged nucleotides. Strikingly, necrotic debris also render macropinocytic, but not non-macropinocytic, pancreas and breast cancer cells resistant to these treatments. Selective, genetic inhibition of macropinocytosis confirms that necrocytosis both supports tumor growth and limits the effectiveness of 5-FU in vivo. Therefore, this study establishes necrocytosis as a mechanism for drug resistance.

[1] Department of Developmental and Cell Biology, University of California Irvine, Irvine, CA 92697, USA. ✉email: aedinger@uci.edu

Because oncogenic mutations constitutively drive anabolism, cancer cells require continuous access to extracellular nutrients for their survival[1]. At the same time, nutrient delivery to tumor cells is limited by abnormal, leaky vasculature, and high interstitial pressure that collapses blood vessels, further compromising perfusion. These limitations make tumor cells reliant on the catabolic process of autophagy[2]. While macroscopic tumor growth depends on autophagy because it keeps tumor cells alive, tumor cell catabolism produces atrophy, not growth. For nutrient-deprived tumor cells to proliferate, they must complement nutrient recycling via autophagy with the scavenging of macromolecules from the microenvironment[1]. Because scavenged nutrients are derived from extrinsic, rather than intrinsic, macromolecules, they can support proliferation as well as survival[3]. Macropinocytosis is one scavenging strategy[1]. Macropinosomes form when plasma membrane ruffles close on themselves and pinch off, producing large, uncoated intracellular vesicles encapsulating extracellular proteins, fluid, and small particles. Oncogenic mutations in RAS or activation of the phosphoinositide 3-kinase (PI3 kinase) pathway can drive macropinocytosis[3–5]. The RAC1 activation required for ruffling can occur via phosphatidylinositol-(3,4,5)-trisphosphate (PIP₃) dependent guanine nucleotide exchange factors[6], downstream of AMP-sensitive kinase (AMPK)[3], or through alternative mechanisms. PIP₃ is also required for macropinosome closure[7]. Pancreas and prostate cancers bearing oncogenic mutations in KRAS or PTEN, respectively, use amino acids derived from engulfed extracellular proteins to proliferate in nutrient-limiting environments[3,4,8–10]. PI3 kinase pathway mutations are also common in breast cancer[11]. While macropinocytosis has not yet been shown to support breast cancer anabolism, this cancer class would likely benefit from macropinocytosis. Desmoplasia, excessive fibrosis that limits perfusion, creates a selective pressure that would favor the outgrowth of breast cancer cells that are capable of nutrient scavenging[12]. Necrosis is a common feature of invasive breast cancer; breast tumor growth often outstrips the vasculature leaving tumor cells nutrient-limited[13]. Necrotic cell debris consumed via macropinocytosis (necrocytosis) could sustain tumor cell anabolism in poorly perfused areas where nutrients are limiting[3].

Many standard-of-care chemotherapeutics kill tumor cells by creating nutrient stress[14]. Some of these agents target enzymes required for de novo nucleotide synthesis, while others cause DNA damage that increases the demand for nucleotides for DNA repair[15]. Autophagy can supply cells with recycled nucleotides and confers resistance to both chemotherapy and radiation[16]. Amino acids provided by macropinocytosis might also confer resistance to therapies that increase the demand for nucleotides by fueling de novo nucleotide synthesis pathways. If macropinocytic cells could directly scavenge nucleotides, even greater protection might be observed because the energetic cost of nucleotide synthesis would be avoided. Necrocytosis could in principle supply the end-products of all biosynthetic pathways. Because nutrients supplied by necrocytosis would be derived from cell-extrinsic sources rather than catabolism, they might allow cells to proliferate in the presence of standard-of-care chemotherapies that target biosynthetic pathways.

Here we demonstrate that breast cancer cell lines with oncogenic mutations that activate KRAS or the PI3K pathway fuel proliferation in nutrient-limiting conditions using not just amino acids, but also sugars, lipids, and nucleotides scavenged via macropinocytosis. Moreover, necrocytosis affords dramatic resistance to a range of standard-of-care therapies that target the metabolic dependencies of cancer cells; albumin supplementation does not provide similar protection. Finally, the use of a genetic strategy that more selectively disables macropinocytosis in tumor cells reveals that macropinocytosis drives both tumor growth and drug resistance in vivo.

## Results

**Macropinocytosis supports anabolism in breast cancer cells.** Although growth factors can induce macropinocytosis in breast cancer cells[17], it was unclear whether breast cancer cells are constitutively macropinocytic or whether macropinocytosis can support breast cancer proliferation when nutrients are limiting. To begin to address these questions, 70 kD dextran uptake was measured in a panel of breast cancer cell lines with activating mutations in KRAS or PIK3CA or with PTEN loss. Because nutrient stress is required to induce macropinocytosis in some cells[3], experiments were conducted in both complete and nutrient-deficient medium containing 1% the normal level of amino acids and glucose (1% AA/gluc). EIPA (5-[N-ethyl-N-isopropyl] amiloride), a Na⁺/H⁺ exchanger (NHE) inhibitor that blocks macropinocytosis but not receptor-mediated endocytosis[3,18], was used to confirm that dextran uptake occurred via macropinocytosis. Immortalized but non-transformed hTERT-HME1 mammary epithelial cells and MCF10A cells did not exhibit macropinocytosis in complete medium, but dextran uptake was stimulated by nutrient deprivation (Fig. 1a and Supplementary Fig. 1a). Similar to pancreas, bladder, colorectal, and lung cancer cell lines with RAS mutations[3,4,19], KRAS-mutant MDA-MB-231 breast cancer cells were robustly macropinocytic in complete medium (Fig. 1a and Supplementary Fig. 1a). MCF-7 and T-47D cells with PIK3CA^E545K and PIK3CA^H1047R mutations, respectively, also efficiently took up high molecular weight dextran in both complete and 1% AA/gluc medium. Indeed, although PIK3CB was found to be required for growth factor-stimulated macropinocytosis[17], oncogenic mutations in PIK3CA were sufficient to induce constitutive macropinocytosis in murine embryonic fibroblasts (MEFs) and non-transformed MCF10A cells[20] (Supplementary Fig. 1b, c) confirming that PI3Kα activation can drive macropinocytosis. Hs578T breast cancer cells carry a mutation in the PI3K regulatory subunit p85α, PIK3R1, that leads to hyper-activation of the PI3K pathway. Like MCF-7 and T-47D cells, Hs578T cells exhibited constitutive macropinocytosis (Fig. 1a and Supplementary Fig. 1a). PTEN-null BT-459 breast cancer cells exhibited constitutive macropinocytosis similar to PTEN-deficient prostate cancer cells[3], while PTEN-deficient MDA-MB-468 cells were contextually macropinocytic in low-nutrient medium (Fig. 1a and Supplementary Fig. 1a). In contrast, PTEN-null HCC1569 cells did not exhibit macropinocytosis even under nutrient stress, although they were capable of performing macropinocytosis when stimulated with phorbol 12-myristate 13-acetate (PMA), an inducer of robust macropinocytosis in multiple cell types (Fig. 1a and Supplementary Fig. 1a, d). 4T1 murine mammary carcinoma cells, a commonly used model for triple-negative breast cancer[21], were also evaluated. While not macropinocytic in complete medium, glucose restriction or direct activation of AMPK with A769662 stimulated macropinocytosis in 4T1 cells similar to results in human MDA-MB-468 triple-negative breast cancer cells (Fig. 1a and Supplementary Fig. 1a, e) and Pten-deficient MEFs[3]. EIPA-sensitive 70 kD FITC-Ficoll uptake was observed in orthotopic, syngeneic 4T1 tumors in female BALB/c mice indicating that AMPK activation or other signals are sufficient to trigger macropinosome formation in vivo (Fig. 1b)[22]. Together, these results (Fig. 1a, b and Supplementary Fig. 1a–e) suggest that many breast tumors are macropinocytic.

Prostate and pancreas cancer cell lines can fuel proliferation in amino acid-limiting conditions with macropinocytosis[3,4,9]. In most studies that measure the anabolic value of macropinocytosis, albumin is provided as a macropinocytic fuel source because it is one of the most abundant extracellular proteins in tumors. However, physiological levels of bovine serum albumin (BSA, 5%) did not support the proliferation of constitutively macropinocytic MCF-7 or T-47D cells or non-macropinocytic HCC1569 cells in medium

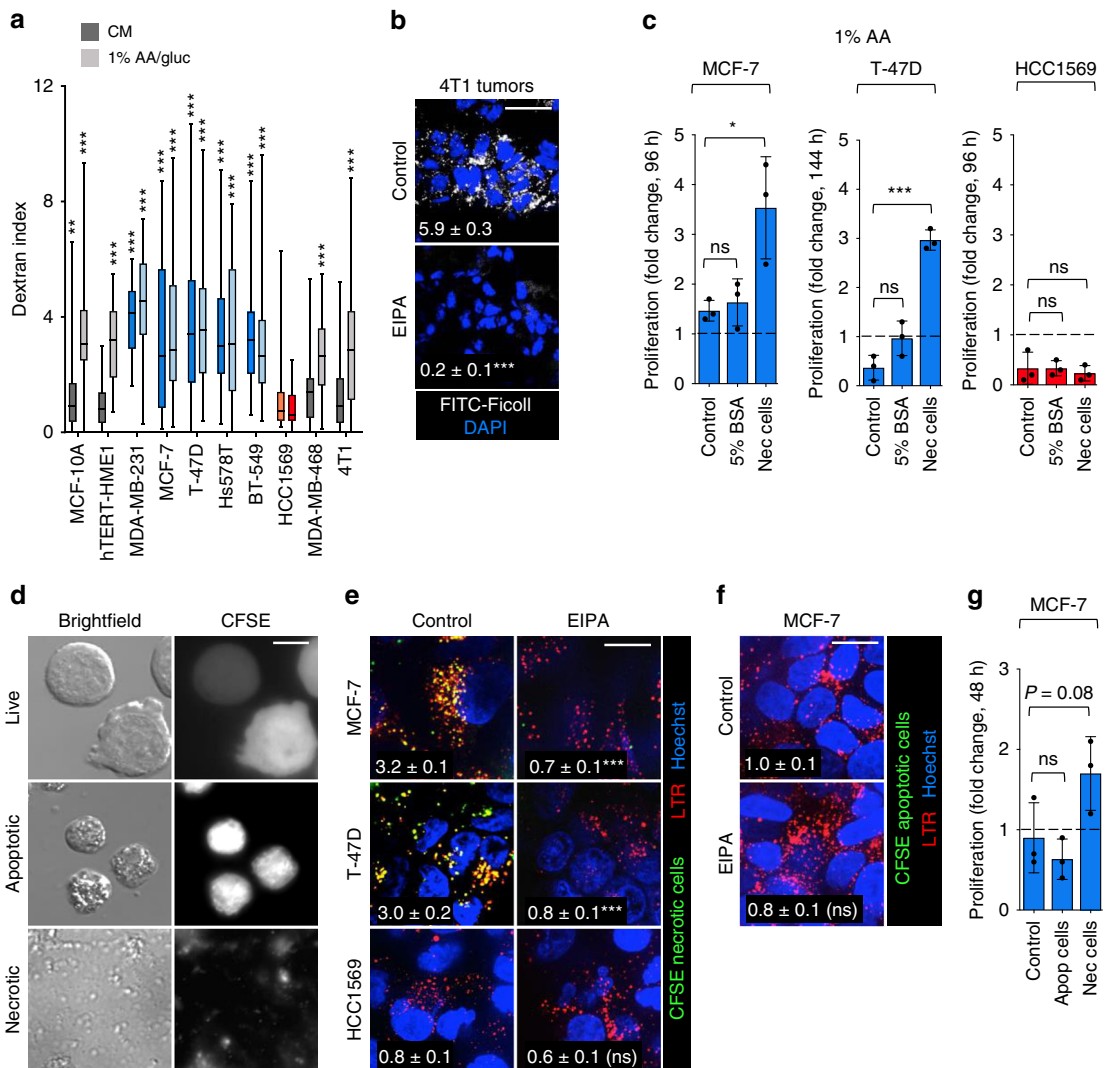

**Fig. 1 Macropinocytosis supports proliferation in nutrient-deprived breast cancer cells. a** Dextran index in complete medium (CM) or 1% AA/gluc medium. Gray (contextual), blue (constitutive), and red (non-macropinocytic) cell lines. Statistics compare cells ± EIPA (50 μM). Using an unpaired, two-tailed $t$-test, ***$P \leq 0.001$; no asterisk, $P > 0.05$; 40 cells were evaluated from $n = 3$ except for MCF10A where $n = 2$. In box plots, the center line is the median, the box is delimited by the 25th and 75th percentile, and whiskers represent minimum and maximum values. **b** FITC-Ficoll uptake in orthotopic 4T1 tumors ± EIPA (10 mg/kg). Ficoll field index calculated from 15 fields from four tumor sections; shown in white. Using an unpaired, two-tailed $t$ test, ***$P \leq$ 0.001. **c** Proliferation of macropinocytic MCF-7 and T-47D cells or non-macropinocytic HCC1569 cells in 1% AA medium ± albumin (5%) or necrotic debris (0.2% protein). Mean ± SD, $n = 3$. Using a one-way ANOVA and Dunnett's correction, *$P \leq 0.05$, ***$P \leq 0.001$, ns, not significant $P > 0.05$. **d** IL-3 was withdrawn from CFSE-labeled FL5.12 cells for 24 or 72 h to produce apoptotic cells or necrotic cell debris, respectively. **e** Uptake of CFSE-labeled necrotic debris in 1% AA/gluc ± EIPA (50 μM). **f** As in **e** but with CFSE-labeled apoptotic cells. In **e**, **f** cells were also stained with LysoTracker Red (LTR). Percent of cell area positive for CFSE in white. Using an unpaired, two-tailed $t$-test, ***$P \leq 0.001$; ns, not significant, $P > 0.05$; $n = 40$ cells from two independent experiments. **g** MCF-7 cell proliferation in 1% AA ± apoptotic cells (1% protein) or necrotic cell debris (0.2% protein). Proliferation was evaluated at 48 h to avoid secondary necrosis of apoptotic cells. Mean ± SD, $n = 3$. Using a one-way ANOVA and Dunnett's correction, ns, not significant, $P > 0.05$. Scale bars, 10 μm (**d**) or 20 μm (**b**, **e** and **f**).

containing 1% of the normal amount of amino acids (1% AA, Fig. 1c). BSA can only provide amino acids at the ratio they are present in its primary sequence, and these levels may not be ideal to support global protein synthesis. Necrotic cell debris is small enough to be consumed by macropinocytosis, a process we have labeled necrocytosis[3]. Upon digestion in the lysosome, dead cell fragments will provide amino acids at the ratio they are present in cellular proteins. As expected, breast cancer cells were able to consume CFSE-labeled necrotic debris, but not larger diameter apoptotic cells (Fig. 1d–f). Necrotic cell debris supported the proliferation of macropinocytic MCF-7 and T-47D breast cancer cells in 1% AA medium even when provided at a 25-fold lower concentration than BSA (0.2% protein, Fig. 1c). Non-macropinocytic HCC1569 cells did

not benefit from supplementation with necrotic debris. Moreover, apoptotic cells that are too large to be consumed by macropinocytosis did not support the proliferation of macropinocytic MCF-7 breast cancer cells under similar conditions (Fig. 1f, g). These studies confirm that the rate of macropinocytic flux in amino acid-limited breast cancer cells is sufficient to support proliferation, that necrotic cell debris is a superior fuel source compared to albumin, and that the nutrients contained in necrotic debris are only accessible to macropinocytic cells.

**Measuring macropinocytic flux using click chemistry.** For macropinocytosed proteins to support protein synthesis in a

nutrient-deprived cell, four processing steps are required: uptake, trafficking to the lysosome, lysosomal proteolysis, and export of the monomeric amino acids to the cytosol[1]. The transfer of isotopically labeled amino acids from the necrotic cells' proteome into the macropinocytic cells' proteome requires that each of these steps is completed and is therefore a holistic measurement of macropinocytic flux[3]. Although highly specific and quantitative, this approach requires relatively expensive labeling medium and proteomics capabilities that are not readily accessible to all laboratories. Copper-catalyzed azido-alkyne cycloaddition, a common form of "click" chemistry[23], offers an alternative strategy for measuring macropinocytic flux (Fig. 2a). A "clickable" alkynyl form of methionine, homopropargylglycine (HPG), is incorporated into proteins, commercially available, and inexpensive[24]. HPG-labeled proteins are readily visualized with Alexa488-streptavidin after clicking on azido-biotin (Fig. 2b, c and Supplementary Fig. 2a).

To validate this strategy for measuring macropinocytic flux (Fig. 2a), HPG-labeled FL5.12 murine hematopoietic cells were generated (Fig. 2b). FL5.12 cells are ideally suited for creating labeled, necrotic debris due to their rapid growth (doubling time of 12 h) and their strict dependence on exogenous IL-3 for survival which allows for the induction of apoptosis and secondary necrosis without the application of noxious chemicals. While adding free HPG to amino acid-deprived MCF-7 (macropinocytic) or HCC1569 (non-macropinocytic) cells resulted in robust labeling of cytosolic and nuclear proteins in both cell types (Fig. 2c), only macropinocytic MCF-7 cells were labeled when HPG was provided in the form of labeled necrotic debris (Fig. 2d–f). At 1 h, HPG labeling was confined to macropinosome-like structures distributed throughout the cytosol (Fig. 2d). Confirming that necrotic debris was consumed via macropinocytosis, the HPG-positive structures in MCF-7 cells were eliminated by EIPA, and no HPG labeling was observed in non-macropinocytic HCC1569 cells incubated with HPG-labeled necrotic debris (Fig. 2d, e). After 24 h, the HPG signal in MCF-7 cells was distributed throughout the nucleus and cytosol, producing a pattern similar to that seen in MCF-7 cells labeled with free HPG (Fig. 2c, g). Similar results were obtained in DU145 prostate cancer cells previously shown to scavenge amino acids via necrocytosis (ref. [3] and Supplementary Fig. 2a–c). Blocking macropinocytosis with EIPA prevented the transfer of HPG from necrotic cell debris into the macropinocytic MCF-7 proteome (Fig. 2g, h). Blocking protein synthesis with cycloheximide resulted in a diffuse cytosolic staining pattern in MCF-7 and DU145 cells fed HPG-labeled necrotic debris confirming that new protein synthesis was required for HPG labeling of nuclear and cytosolic proteins in macropinocytic cells (Fig. 2g, h and Supplementary Fig. 2b, c). Non-macropinocytic HCC1569 breast cancer cells did not incorporate HPG from necrotic debris even when the incubation period was extended to 24 h (Fig. 2h, i). In summary, a clickable amino acid tracer can be used to monitor macropinocytic flux and confirmed that nutrient-deprived, macropinocytic breast cancer cells can fuel new protein synthesis with proteins scavenged from dead cell corpses.

**Necrocytosis provides access to carbohydrates.** Several studies have demonstrated that amino acids can be scavenged from macropinocytosed proteins[3,4,8,9,25]. However, provided that a macromolecule can be broken down in the lysosome and the subunits exported to the cytosol, necrotic cell debris contains all of the building blocks necessary to produce a new cell. Cancer cells import and oxidize glucose to generate ATP, but glucose is also funneled into the hexosamine biosynthesis pathway to produce N-acetylglucosamine (GlcNAc) required for protein

O-GlcNAcylation reactions[26]. To assess whether GlcNAc, and potentially other carbohydrates, can be scavenged via necrocytosis, FL5.12 cells were labeled with a clickable, alkynyl form of GlcNAc, tetraacylated N-(4-pentynoyl)-glucosamine (Ac4GlcNAlk) (Supplementary Fig. 3a)[27]. The acetyl groups on Ac4GlcNAlk increase the membrane permeability of this unnatural sugar but are removed by carboxyesterases in the cytosol, generating the monosaccharide. FL5.12, MCF-7, and HCC1569 cells all labeled efficiently with free Ac4GlcNAlk, producing labeling patterns consistent with the wide range of membrane, cytosolic, and nuclear proteins that undergo O-GlcNAcylation (Supplementary Fig. 3a, b)[28]. Macropinocytic MCF-7 breast cancer cells, but not non-macropinocytic HCC1569 cells, were able to recover GlcNAlk from labeled necrotic cell debris, producing a staining pattern similar to that observed upon free Ac4GlcNAlk addition (Supplementary Fig. 3b–e). Blocking macropinocytosis with EIPA significantly reduced GlcNAlk flux from necrotic debris into MCF-7 cells (Supplementary Fig. 3c, d). As EIPA completely blocked HPG incorporation from labeled necrotic debris under similar experimental conditions (Fig. 2g, h), MCF-7 cells may secrete enzymes that liberate free GlcNAlk from necrotic debris. Because non-macropinocytic HCC1569 breast cancer cells did not label when incubated with GlcNAlk-labeled necrotic cell debris (Supplementary Fig. 3d, e), enzymes that release free GlcNAlk are not present in the necrotic debris itself. The ability to scavenge carbohydrates may contribute to the ability of macropinocytic MCF-7 and T-47D, but not non-macropinocytic HCC1569, cells to proliferate in medium deficient in both amino acids and glucose (1% AA/gluc) in the presence of necrotic debris (Supplementary Fig. 3f, g). As expected, blocking necrocytosis with EIPA eliminated this proliferation. In summary, sugars can be scavenged from necrotic material via macropinocytosis.

**Necrocytosis relieves dependence on fatty acid synthesis.** While necrocytosis preserves lipid droplets in nutrient-restricted prostate cancer cells[3], it was not clear whether lipids can be directly scavenged from cell corpses or whether scavenging of other nutrients simply reduces ATP demand and, consequently, lipid catabolism in the macropinocytic cell. To directly monitor lipid flux from necrotic cell debris into macropinocytic breast cancer cells, FL5.12 cells were labeled with alkynyl palmitate, a fatty acid used for fatty acid oxidation, post-translational protein modifications, and to build cell membranes[29]. FL5.12, MCF-7, and HCC1569 cells labeled efficiently with free alkynyl palmitate (Fig. 3a, b). Similar to results with HPG and GlcNAlk (Fig. 2 and Supplementary Fig. 3), macropinocytic MCF-7, but not non-macropinocytic HCC1569, breast cancer cells recovered alkynyl palmitate from labeled necrotic debris (Fig. 3c, d and Supplementary Fig. 4a). Inhibiting macropinocytosis with EIPA eliminated labeling in MCF-7 cells confirming that macropinocytosis was required. Thus, necrocytosis provides macropinocytic breast cancer cells with fatty acids at what should be physiologic ratios.

Fatty acid synthesis is critical for tumor cell growth[14,30]. Fatty acid synthase (FASN) is over-expressed in some breast tumors and FASN inhibition limits breast tumor growth[31]. As macropinocytic breast cancers can scavenge fatty acids from necrotic cell debris (Fig. 3c, d), necrocytosis may reduce dependence on FASN and thus sensitivity to FASN inhibitors (FASNi). The FASNi GSK2194069[32] killed MCF-7 cells and slowed the proliferation of HCC1569 breast cancer cells (Fig. 3e, f). Consistent with the proposal that fatty acid scavenging would reduce dependence on FASN, supplementation with necrotic cell debris rescued macropinocytic MCF-7, but not non-macropinocytic HCC1569, breast cancer cells from GSK2194069. While MCF-7 cells were also rescued from

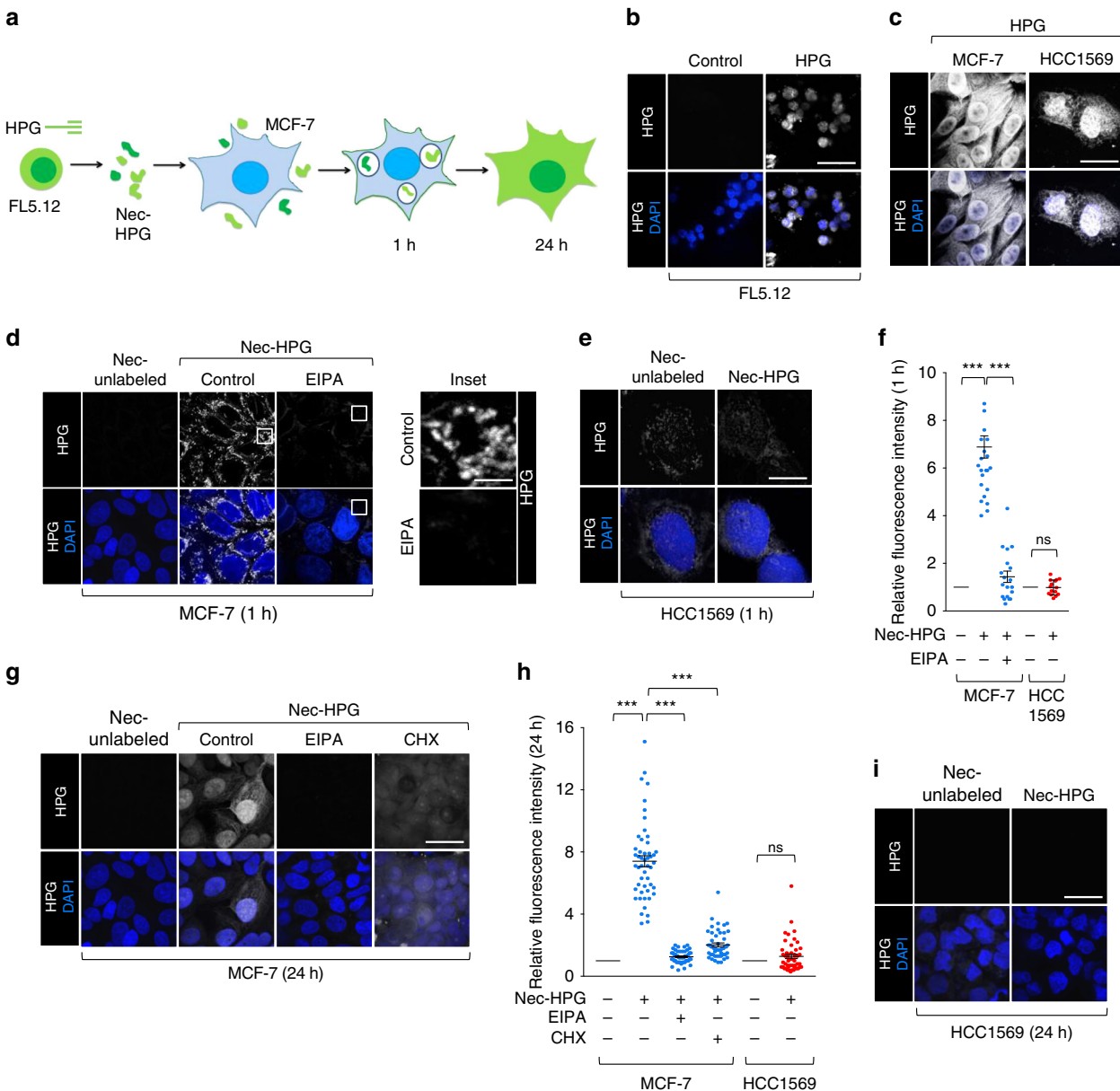

**Fig. 2 Measurement of macropinocytic protein flux in breast cancer cells. a** Assay schematic. FL5.12 cells were labeled with HPG prior to the induction of necrosis. HPG-labeled necrotic debris was added to macropinocytic MCF7 cells for 1 or 24 h. HPG incorporation into the MCF-7 cell proteome was detected with clickable biotin-azide followed by streptavidin-Alexa488. **b** FL5.12 cells were labeled with HPG for 24 h in complete medium. **c** MCF-7 or HCC1569 cells labeled as in **b** but in 1% AA medium. **d, e** MCF-7 (**d**) or HCC1569 (**e**) cells in 1% AA medium were supplemented with unlabeled or HPG-labeled necrotic cell debris (nec-HPG) for 1 h. MCF-7 cells ± EIPA (50 μM). **f** Integrated fluorescence intensity per cell in **d, e** normalized to cells fed unlabeled necrotic debris. A total of 25 cells (MCF-7) or 20 cells (HCC1569) from one biological replicate. **g** As in **d** but evaluated at 24 h ± EIPA or ± cyclohexamide (50 μg/ml). **h** Integrated fluorescence intensity per cell from (**g, i**) normalized to cells fed unlabeled necrotic debris; 50 cells were evaluated from $n = 3$. **i** HCC1569 cells were supplemented with necrotic cell debris as in **e** but for 24 h. Scale bars, 20 μm. In **f, h** using a one-way ANOVA and Tukey's correction (MCF-7) or an unpaired, two-tailed t-test (HCC1569), ***$P \le 0.001$; ns, not significant, $P > 0.05$; mean ± SEM shown.

GSK2194069 by supplementation with free fatty acids as expected, HCC1569 cells were not (Supplementary Fig. 4b). However, contextually macropinocytic MDA-MB-468 cells were rescued from FASN inhibition by free fatty acids, but not necrotic debris, validating this approach (Supplementary Fig. 4c, d). Prostate cancers both over-express FASN and depend on fatty acid uptake[33]; blocking either process limits prostate cancer cell growth and survival. *PTEN*-deficient prostate cancers (e.g., LNCaP) are macropinocytic[3], while *PTEN*-replete 22Rv1 prostate cancer cells are not (Fig. 3g). Both LNCaP and 22Rv1 prostate cancer cells died when exposed to the FASNi GSK2194069 and

were rescued by free fatty acids (Fig. 3h, i and Supplementary Fig. 4e, f). Similar to results with breast cancer cells, macropinocytic LNCaP cells, but not non-macropinocytic 22Rv1 cells, were also rescued from FASN inhibition by supplementation with necrotic cell debris (Fig. 3h, i). These results demonstrate that necrocytosis provides fatty acids and affords resistance to therapeutics that limit lipid biosynthesis.

**Necrocytosis can supplant nucleotide biosynthesis.** Nucleotide synthesis represents a metabolic bottleneck for rapidly proliferating

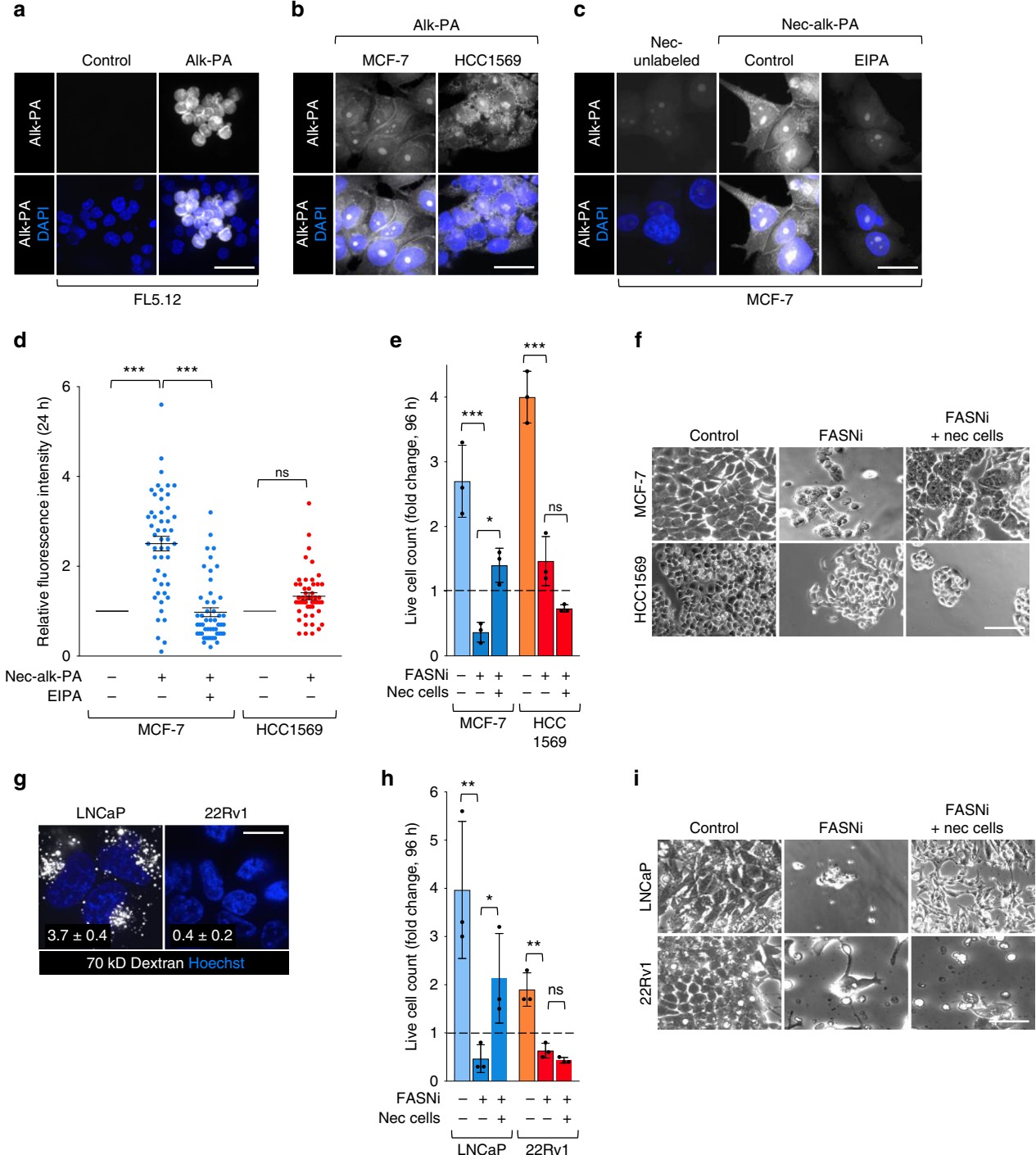

**Fig. 3 Necrocytosis supplies fatty acids and confers resistance to fatty acid synthase inhibition. a** FL5.12 cells labeled with alkynyl-PA in complete medium for 24 h; alk-PA detected with biotin-azide and streptavidin-Alexa488. **b** MCF-7 and HCC1569 cells labeled as in **a**. **c** Macropinocytic MCF-7 cells maintained for 24 h ± EIPA (50 μM) and supplemented with unlabeled or alk-PA-labeled necrotic cell debris (nec-alk-PA). **d** Integrated fluorescence intensity per cell from (**c**, or Supplementary Fig. 4a) normalized to cells fed unlabeled necrotic cell debris. A total of 50 cells were quantified from two independent experiments. Using a one-way ANOVA and Tukey's correction (MCF-7) or an unpaired, two-tailed t-test (HCC1569); ***$P \leq 0.001$, ns, not significant $P > 0.05$. **e** Proliferation of macropinocytic MCF-7 or non-macropinocytic HCC1569 cells ± fatty acid synthase inhibitor (FASNi) (GSK2194069, 20 μM) ± necrotic debris (0.2% protein) at 96 h. **f** Representative bright field images for proliferation assay in **e**. **g** 70 kD dextran uptake in LNCaP or 22Rv1 prostate cancer cells. Mean dextran index ± SEM shown in white; 50 cells were quantified from 1 (22Rv1) or 3 (LNCaP) independent experiments. **h** Proliferation of macropinocytic LNCaP cells or non-macropinocytic 22Rv1 cells ± FASNi (GSK2194069, 20 μM) ± necrotic debris (0.2% protein) at 96 h. **i** Representative bright field images for proliferation assay in **h**. For **e**, **h** mean ± SD, $n = 3$. Using a one-way ANOVA and Tukey's correction, *$P \leq 0.05$; **$P \leq 0.01$; ***$P \leq 0.001$; ns, not significant, $P > 0.05$. Scale bars, 20 μm.

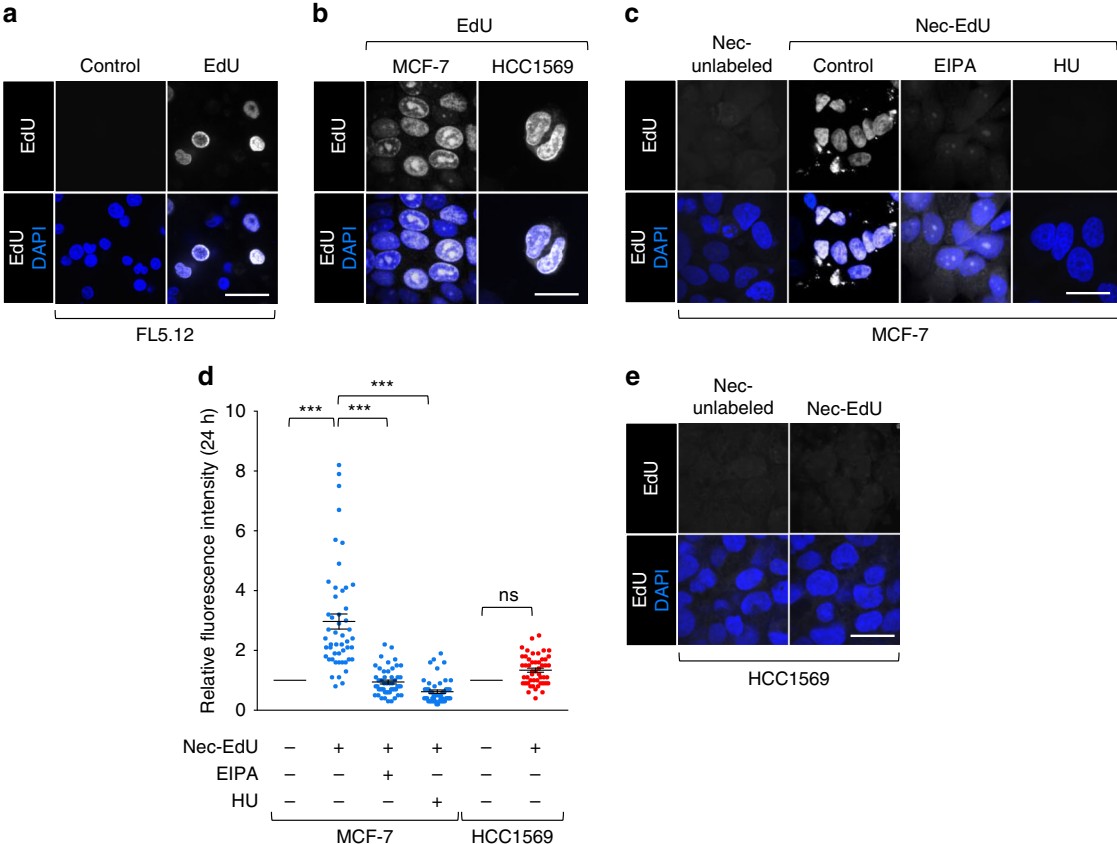

**Fig. 4 Necrocytosis provides nucleotides. a** FL5.12 cells labeled with EdU in complete medium; EdU detected with biotin-azide and streptavidin-Alexa488. **b** MCF-7 and HCC1569 cells labeled with free EdU as in **a**. **c** Macropinocytic MCF-7 cells ± EIPA (50 μM) or hydroxyurea (HU, 10 mM) were supplemented with unlabeled or EdU-labeled necrotic cell debris (nec-EdU). **d** Integrated fluorescence intensity per cell in **c**, **e** normalized to cells fed unlabeled necrotic cell debris. A total of 50 cells were quantified from three independent experiments except for hydroxyurea treatment where $n = 2$; mean ± SEM shown. Using a one-way ANOVA and Tukey's correction (MCF-7) or an unpaired, two-tailed $t$-test (HCC1569), ***$P \leq 0.001$; ns, not significant, $P > 0.05$. **e** As in **c** but for non-macropinocytic HCC1569 cells. Scale bars, 20 μm.

cancer cells[14]. Amino acids derived from macropinocytosed proteins could be used to support nucleotide synthesis. Alternatively, directly scavenging nucleotides via necrocytosis would avoid energetically demanding nucleotide biosynthesis. When the alkynyl thymidine analog 5-ethynyl-2-deoxyuridine (EdU)[34] was added to the culture medium, it was readily incorporated into the genomic DNA of FL5.12, MCF-7, and HCC1569 cells as expected (Fig. 4a, b). Akin to results obtained with HPG, GlcNAlk, and alkynyl palmitate, macropinocytic MCF-7 cells, but not non-macropinocytic HCC1569 cells, were able to recover EdU from necrotic cell debris (Fig. 4c–e). Blocking macropinocytosis with EIPA prevented the transfer of EdU from necrotic cell debris to MCF-7 cells, again implicating macropinocytosis in this process. Blocking DNA replication with hydroxyurea[35] also eliminated the nuclear EdU signal in MCF-7 cells, confirming that EdU was incorporated into genomic DNA. In sum, macropinocytic cells can recover nucleotides directly from necrotic debris.

Many standard-of-care cancer therapies target nucleotide biosynthesis directly or indirectly[36,37]. Agents such as 5-fluorouracil (5-FU) and gemcitabine block nucleotide biosynthesis, while DNA damaging agents (e.g., doxorubicin or ionizing radiation) increase dependence on nucleotide synthesis due to the need for DNA repair. Many different solid tumors initially respond to genotoxic chemotherapies, but resistance frequently develops. Metabolic adaptations include the up-regulation of de novo nucleotide synthesis pathways, increased nutrient import, and recycling via autophagy[15,38,39]. By providing breast

cells with scavenged nucleotides (Fig. 4), necrocytosis may also contribute to resistance to cancer therapies that deplete nucleotide pools. Breast cancer is treated with 5-FU, a pyrimidine analog that inhibits thymidylate synthase[40]. Both MCF-7 and HCC1569 cells were sensitive to 5-FU (Fig. 5a, b). Supplementation with necrotic cell debris afforded striking protection to 5-FU-treated macropinocytic MCF-7 cells, restoring proliferation to the level seen in untreated cells. As necrotic cell debris did not protect HCC1569 cells, the debris is not simply reducing toxicity by sequestering 5-FU. As expected, EIPA eliminated the ability of necrotic debris to rescue MCF-7 cells from 5-FU, and supplying the thymidine precursor deoxythymidine monophosphate (dTMP) rescued both macropinocytic and non-macropinocytic breast cancer cells from 5-FU (Supplementary Fig. 5a–c). Like 5-FU, gemcitabine blocks nucleotide synthesis, but does so by inhibiting ribonucleotide reductase[40]. Gemcitabine is a standard-of-care treatment for pancreas cancer, a tumor type that is frequently macropinocytic. Gemcitabine killed both macropinocytic PANC-1 and non-macropinocytic BxPC3 cells (Fig. 5c, d). Supplementation with necrotic cell debris fully restored proliferation in PANC-1 cells while failing to benefit BxPC3 cells that are incapable of necrocytosis[3]. Thus, necrocytosis affords dramatic protection from standard-of-care chemotherapies that target nucleotide synthesis pathways.

DNA damaging agents like the topoisomerase inhibitor doxorubicin, increase the demand for nucleotide synthesis[15]. Doxorubicin triggered cell death in MCF-7, MDA-MB-231, and

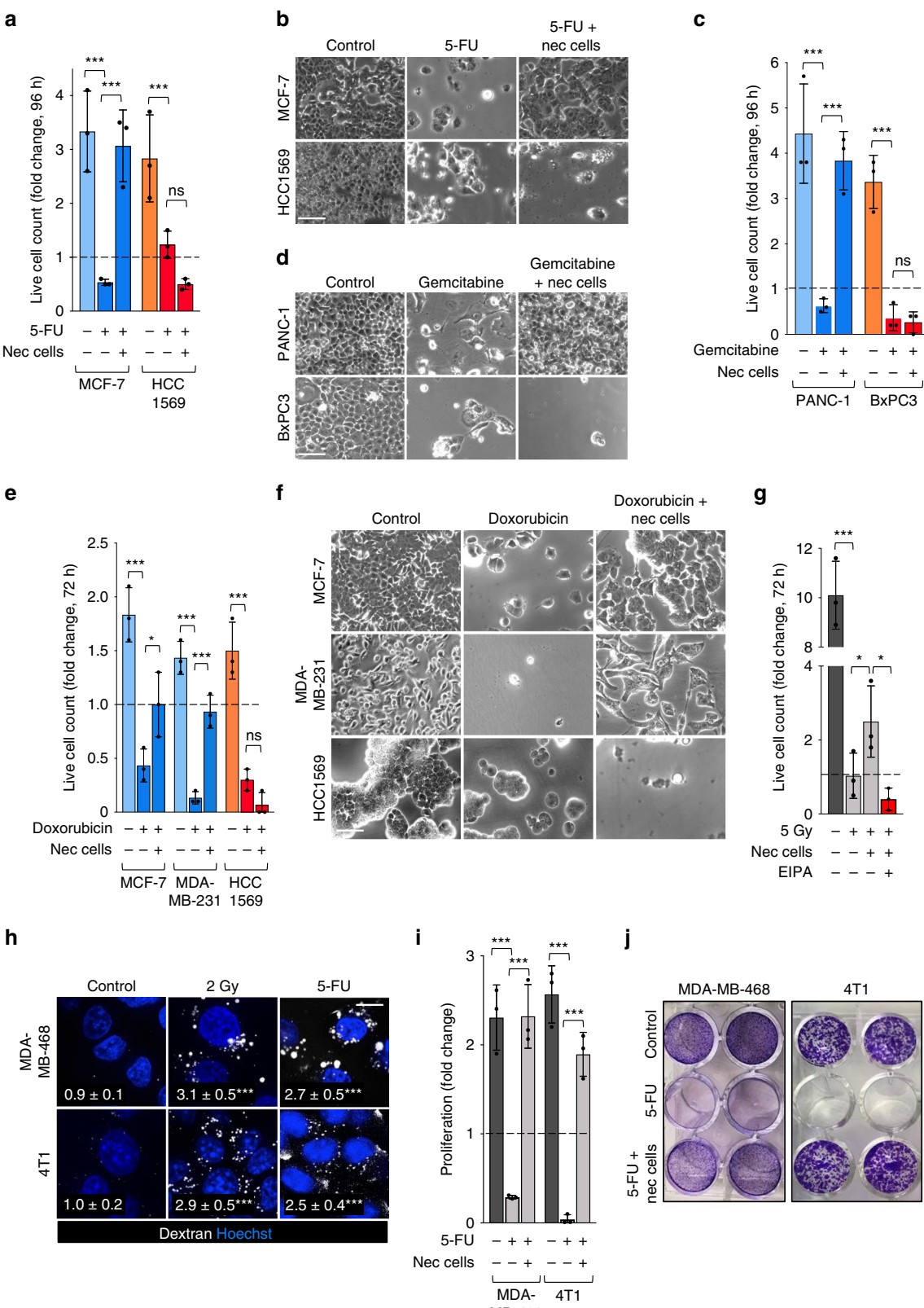

HCC1569 breast cancer cells (Fig. 5e, f). Necrotic cell debris prevented death in both of the macropinocytic cell lines, MCF-7 and MDA-MB-231, while non-macropinocytic HCC1569 cells remained fully sensitive to doxorubicin. Murine 4T1 mammary carcinoma cells were sensitive to γ-irradiation (Fig. 5g). Radiation has been reported to increase AMPK activation[41,42], and

γ-irradiation stimulated macropinocytosis in both 4T1 and MDA-MB-468 cells maintained in complete medium (Fig. 5h). Consistent with the stimulation of macropinocytosis by γ-irradiation (Fig. 5h), supplementation with necrotic cell debris allowed irradiated 4T1 breast cancer cells to proliferate in an EIPA-sensitive manner (Fig. 5g). Interestingly, 5-FU treatment

**Fig. 5 Necrocytosis confers resistance to standard-of-care therapies that target or create dependence on nucleotide biosynthesis. a** Proliferation of macropinocytic MCF-7 or non-macropinocytic HCC1569 cells ± 5-FU (30 μM) ± necrotic debris (0.2% protein) at 96 h. **b** Representative bright field images for proliferation assay in **a**. **c** Proliferation of macropinocytic PANC-1 or non-macropinocytic BxPC-3 cells ± necrotic debris (0.2% protein) ± gemcitabine (20 μM) at 96 h. **d** Representative bright field images for proliferation assay in **c**. **e** Proliferation of macropinocytic MCF-7 and MDA-MB-231 or non-macropinocytic HCC1569 cells ± doxorubicin (1 μM) ± necrotic debris (0.2% protein) at 72 h. **f** Representative bright field images for proliferation assay in **e**. **g** Proliferation of 4T1 cells in CM subjected to 5 Gy of γ-irradiation ± nec cells and ± EIPA (10 μM) at 72 h. **h** Dextran uptake in 4T1 or MDA-MB-468 cells 2 h after γ-irradiation (2 Gy) or 1 h after treatment with 5-FU (5 μM). Statistics compare dextran index (in white) in control and irradiated or treated cells; 30 cells from two independent experiments were evaluated. Using a one-way ANOVA and Dunnett's correction, ***$P \leq 0.001$. **i** Proliferation of 4T1 or MDA-MB-468 cells ± 1 μM (4T1) or 20 μM (MDA-MB-468) 5-FU ± necrotic debris (0.2% protein) at 72 h (4T1) or 96 h (MDA-MB-468). **j** Representative images for **i** (duplicate wells shown). In **a**, **c**, **e**, **g** and **i**, mean ± SD shown, $n = 3$; Using a one-way ANOVA and Tukey's correction, *$P \leq 0.05$; ***$P \leq 0.001$; ns, not significant, $P > 0.05$.

was also sufficient to stimulate macropinocytosis in contextually macropinocytic 4T1 and MDA-MB-468 breast cancer cells (Fig. 5h), and necrotic cell debris rendered both cell lines resistant to 5-FU (Fig. 5i, j). Microtubule-stabilizing agents do not kill cancer cells by inducing metabolic stress. Consistent with this, necrocytosis did not protect macropinocytic LNCaP cells from docetaxel-induced death (Supplementary Fig. 5d, e). Together, these findings demonstrate that necrocytosis can render cancer cells resistant to a variety of metabolic cancer therapies by providing the end products of biosynthesis.

**Macropinocytosis confers resistance to chemotherapy in vivo.** While necrocytosis supported cancer cell proliferation and provided resistance to metabolic chemotherapies in vitro, it was unclear whether viable cells in the tumor microenvironment would have sufficient access to dead cells or other high-quality macropinocytic fuel to drive tumor growth or drug resistance. We first considered whether EIPA could be used to dissect the contribution macropinocytosis makes to tumor growth. However, EIPA completely blocked the proliferation of non-macropinocytic breast, prostate, and pancreas cancer cells in complete medium containing abundant nutrients (Fig. 6a and Supplementary Fig. 6a). These results are consistent with EIPA's significant, macropinocytosis-independent anti-proliferative effects[43]. Selective, genetic approaches have enabled landmark studies dissecting the role of autophagy in tumor initiation and progression[44]. In contrast, no proteins have been identified that function only in macropinocytosis, precluding replacing EIPA with a genetic knockout strategy. However, a double point mutant of the actin capping protein regulator CARMIL1, CARMIL1 KR987/989AA (CARMIL1-AA), supports normal RAC GTPase activation and cell migration, but not macropinocytosis[45]. CARMIL1-AA is not a dominant-negative mutant, and thus the endogenous CARMIL1 protein must be knocked down to observe macropinocytosis inhibition. As expected, reconstituting 4T1 CARMIL1 knockdown cells with an shRNA-resistant cDNA encoding wild type CARMIL1 (CARMIL1-WT), but not mutant CARMIL1-AA, restored macropinocytosis in response to both nutrient deprivation and 5-FU (Fig. 6b, c and Supplementary Fig. 6b). Importantly, CARMIL1-WT and CARMIL1-AA 4T1 cells proliferated equally well in complete medium where neither is macropinocytic (Fig. 6d and Supplementary Fig. 6c). However, CARMIL1-AA cells were unable to use necrocytosis to fuel proliferation in low-nutrient medium or in the presence of 5-FU (Fig. 6d, e and Supplementary Fig. 6c, d). Together, these results demonstrate that replacing endogenous CARMIL1 with the CARMIL1-AA mutant blocks macropinocytosis-driven proliferation more selectively than does the commonly used macropinocytosis inhibitor, EIPA.

CARMIL1-WT and CARMIL1-AA 4T1 cells were employed to measure the contribution macropinocytosis makes to tumor

growth in vivo. Consistent with in vitro assays (Fig. 6b), orthotopic, syngeneic tumors formed from CARMIL1-WT, but not CARMIL1-AA, 4T1 cells were macropinocytic (Fig. 7a). Non-macropinocytic CARMIL1-AA tumors took longer to reach a volume of 100 mm³, and a subset of CARMIL1-AA tumors grew more slowly that macropinocytosis-competent CARMIL1-WT tumors (Fig. 7b, c and Supplementary Fig. 7a). These differences translated into a significant survival benefit for mice bearing CARMIL1-AA tumors (Fig. 7d). 4T1 tumors exhibit rapid growth and were highly necrotic as expected (Supplementary Fig. 7b). Necrotic cells were distributed throughout the tumor suggesting that necrotic debris is readily accessible to viable 4T1 tumor cells. Given that necrotic cell debris, but not 5% albumin, is sufficient to promote 4T1 cell proliferation in low nutrients in vitro (Supplementary Fig. 7c), it is likely that necrocytosis accounts for the more robust growth of CARMIL1-WT tumors in vivo (Fig. 7b–d). Because CARMIL1-AA is more selective than EIPA (Fig. 6a, d and Supplementary Fig. 6a, c), these tumor growth assays also provide more rigorous support for the model that macropinocytosis drives solid tumor growth than published studies using EIPA[3,4,8].

Whether necrocytosis could protect tumors from 5-FU was next assessed. In vitro, macropinocytic CARMIL1-WT, but not non-macropinocytic CARMIL1-AA, 4T1 cells were resistant to 5-FU selectively in the presence of necrotic debris (Fig. 6e and Supplementary Fig. 6d). Importantly, 5% albumin did not rescue from 5-FU (Supplementary Fig. 7d). To evaluate whether necrocytosis provided resistance to chemotherapy in vivo, CARMIL1-WT or CARMIL1-AA tumors were treated with 30 mg/kg 5-FU every 3 days for 3 doses once each tumor reached 100 mm³. During the 5-FU treatment period, the average volume of CARMIL1-AA tumors did not increase while CARMIL1-WT tumors continued to grow (Fig. 7e and Supplementary Fig. 7e). Upon termination of 5-FU treatment, tumor growth resumed, although at a slower rate for 5-FU-treated CARMIL1-AA than CARMIL1-WT tumors. This difference in response resulted in a significantly extended survival time in 5-FU-treated mice bearing CARMIL1-AA tumors relative to both 5-FU-treated mice bearing CARMIL1-WT tumors and to vehicle-treated mice bearing CARMIL1-AA tumors (Fig. 7f). Taken together, these results provide strong evidence that necrocytosis can render macropinocytic tumor cells resistant to standard-of-care therapies that target biosynthetic pathways.

**Discussion**
Many tumor cells are likely macropinocytic. Activation of the RAS and PI3 kinase pathways is common across cancer classes and has been shown to drive macropinocytosis in pancreas, lung, colorectal, bladder, prostate, and now breast cancer cells[3,4,19,46] (Fig. 1a). Importantly, the rate of macropinocytic flux in breast cancer cells is sufficient to support proliferation in nutrient-

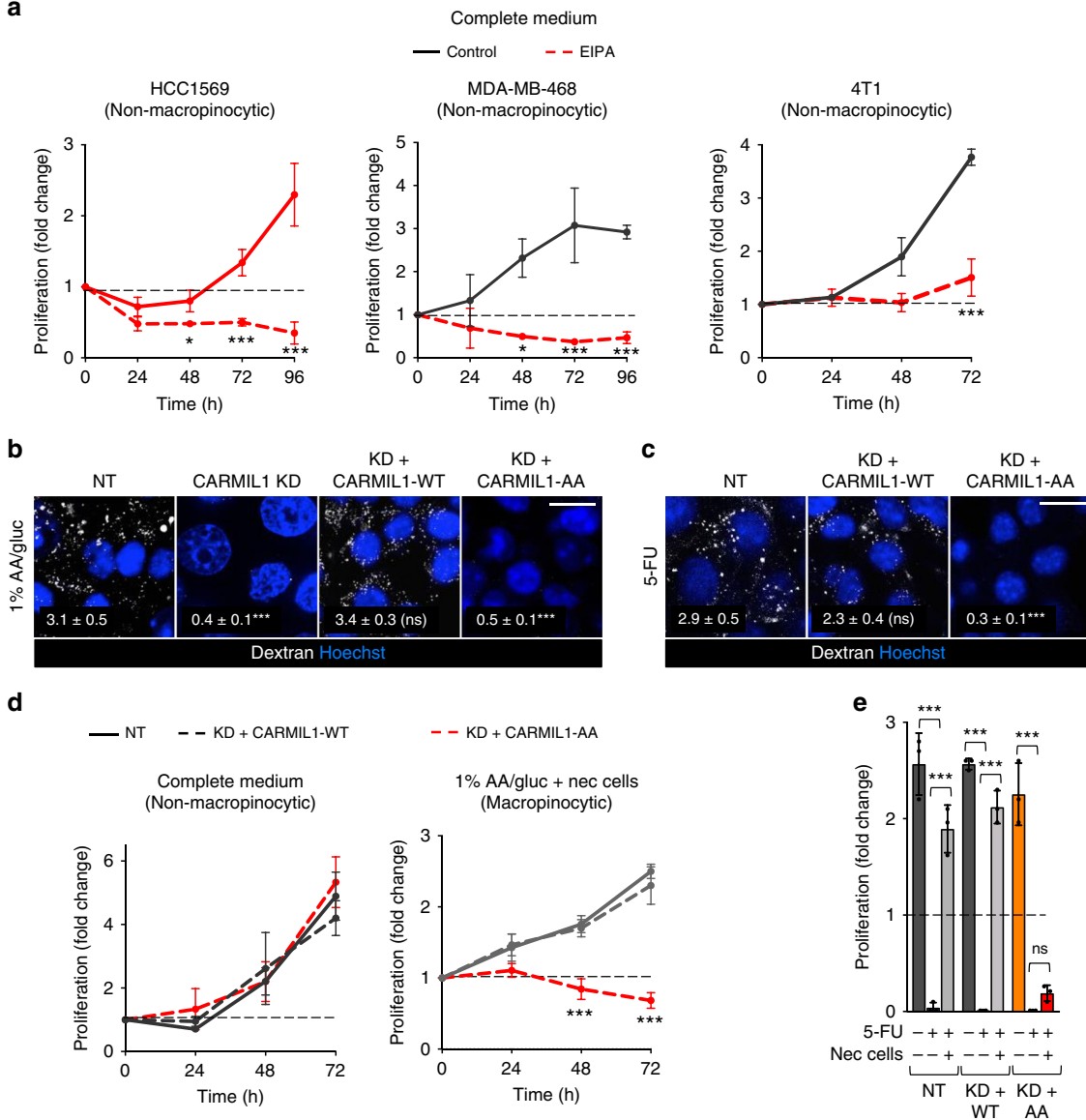

**Fig. 6 CARMIL1-AA supports proliferation, but not macropinocytosis, in mammary cancer cells. a** Breast cancer cell line proliferation in complete medium ± EIPA (25 μM (HCC1569 and MDA-MB-468) or 10 μM (4T1, 4T1 cells are hypersensitive to EIPA)). Mean ± SD shown, $n = 3$. Using an unpaired, two-tailed t-test at each time point, $*P \leq 0.05$; $***P \leq 0.001$; no asterisk, $P > 0.05$. **b** Dextran uptake in 4T1 cells expressing a non-targeting shRNA (NT) or a CARMIL1-shRNA in 1% AA/glucose. Where indicated, CARMIL1 knockdown cells were reconstituted with shRNA-resistant CARMIL1-WT or CARMIL1-AA cDNAs. **c** As in **b**, but in complete medium 1 h after treatment with 5-FU (5 μM). For **b** and **c**, dextran index in white (mean ± SEM); 30 cells from two independent experiments were evaluated. Using a one-way ANOVA with Dunnett's correction, $***P \leq 0.001$; ns, not significant, $P > 0.05$. Scale bars, 20 μm. **d** Proliferation of 4T1 cells in complete medium or 1% AA/gluc with necrotic cells (0.2% protein). Mean ± SD shown, $n = 3$. Using a one-way ANOVA at each time point and Dunnett's correction, $***P \leq 0.001$; no asterisk, $P > 0.05$. **e** As in **d**, but cells in complete medium were treated with 5-FU (1 μM) ± necrotic debris (0.2% protein) and proliferation measured at 72 h. Mean ± SD shown, $n = 3$. Using a one-way ANOVA and Tukey's correction, $***P \leq 0.001$; ns, not significant, $P > 0.05$.

limiting conditions provided that necrotic corpses are supplied as fuel (Fig. 1c and Supplementary Fig. 3f). Albumin can support the proliferation of some macropinocytic cells in low glutamine or when non-essential amino acids are limiting[4,10], but was not sufficient to support breast cancer cell proliferation in media generally deficient in amino acids, in both amino acids and glucose, or in cells exposed to 5-FU (Fig. 1c and Supplementary Figs. 3f and 7c, d). As necrotic cell debris supported proliferation in these same conditions even when provided at a 25-fold lower concentration than albumin, necrotic cell debris is a much higher quality macropinocytic fuel source. These results highlight that in vitro experiments where only BSA is supplied as fuel are likely

to underestimate the potential contribution of macropinocytosis to tumor cell anabolism. At the same time, it is difficult to determine what a physiologically relevant amount of necrotic cell debris would be. Dead cells are present throughout tumors, not just in large, necrotic foci, and comprise a significant fraction of the tumor volume (e.g., 70–80% in Supplementary Fig. 7b). It has been reported that triggering apoptosis paradoxically promotes tumorigenesis by stimulating the proliferation of neighboring viable cells[47]. This death-driven proliferation may be fueled in part by necrocytosis; when apoptotic cells undergo secondary necrosis, "bite-sized" fragments are created. Co-injection of dead cells with viable 4T1 cells accelerates their proliferation in

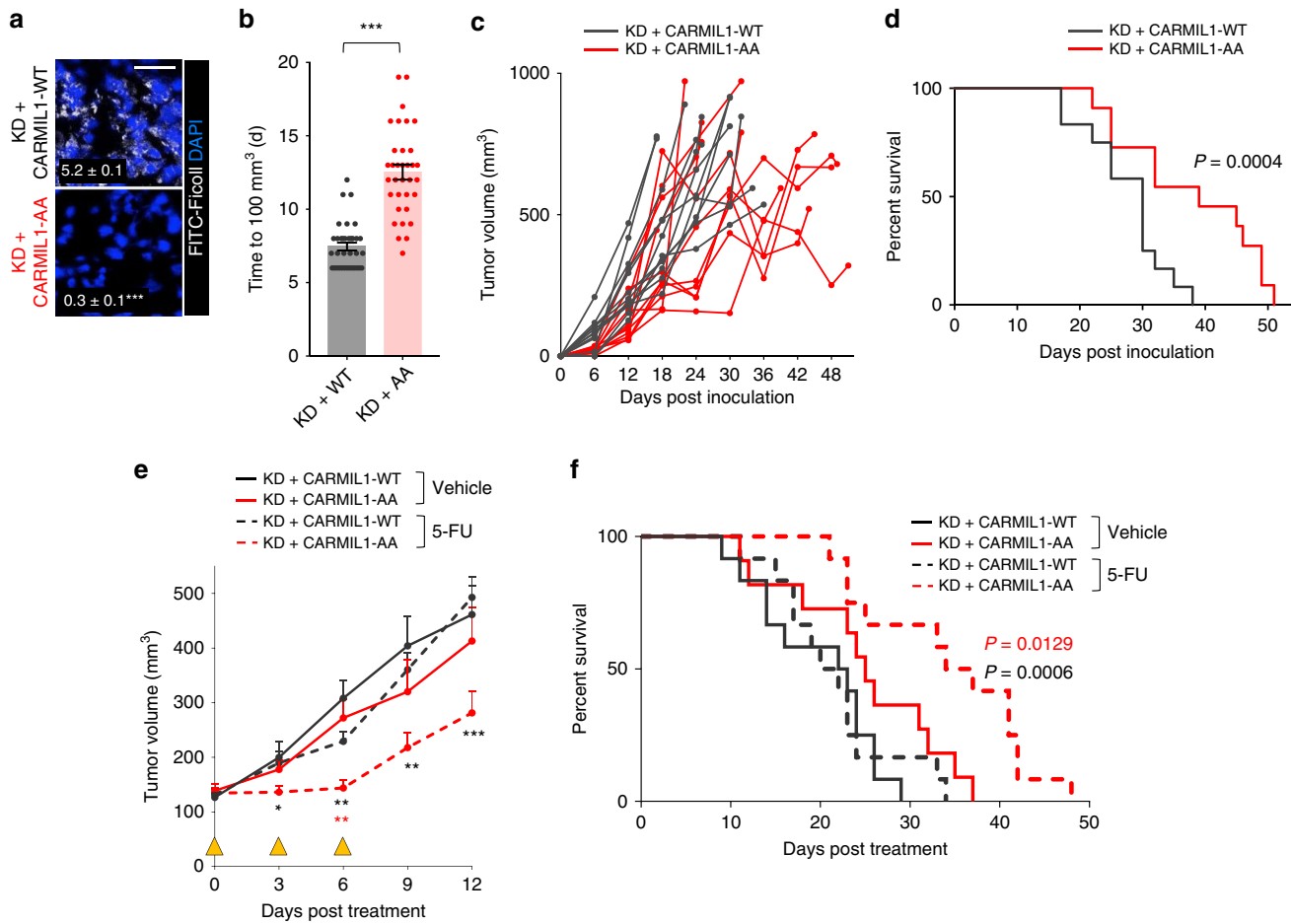

**Fig. 7 Macropinocytosis promotes tumor growth and fuels resistance to 5-FU in vivo. a** FITC-Ficoll uptake in orthotopic 4T1 tumors generated from CARMIL1-WT or CARMIL1-AA cells. Ficoll field-index was calculated from 10 fields from 3 (CARMIL1-WT) or 4 (CARMIL1-AA) tumor sections; Ficoll field index ± SEM in white. Using an unpaired, two-tailed t test, ***$P \leq 0.001$. Scale bar, 20 μm. **b** Time to reach 100 mm³ for tumors formed from CARMIL1-WT ($n = 26$) or CARMIL1-AA cells ($n = 25$); tumor volume was measured with calipers daily starting on day 6. Using an unpaired, two-tailed t test, ***, $P \leq 0.001$, mean ± SEM. **c** Growth of the tumors in **b**. **d** Kaplan–Meier survival analysis in mice bearing CARMIL1-WT ($n = 12$) or CARMIL1-AA ($n = 11$) 4T1 tumors. P-value calculated using a two-sided log-rank (Mantel-Cox) test. **e** As in **b**, but animals were treated with vehicle (CARMIL1-WT, $n = 12$; CARMIL1-AA, $n = 11$) or 30 mg/kg 5-FU (CARMIL1-WT, $n = 12$; CARMIL1-AA, $n = 12$); mean ± SEM. Yellow triangles mark days 5-FU was administered. Statistics compare CARMIL1-WT vs. CARMIL1-AA tumors treated with 5-FU (black asterisks) or CARMIL1-AA tumors treated with vehicle or 5-FU (red asterisks). Using a one-way ANOVA and Tukey's correction, *$P \leq 0.05$; **$P \leq 0.01$; ***$P \leq 0.001$; no asterisk, $P > 0.05$. **f** Kaplan–Meier survival curve beginning after initiation of 5-FU treatment. P-value calculated using a two-sided log-rank (Mantel-Cox) test and compare 5-FU-treated CARMIL1-WT ($n = 12$) and CARMIL1-AA tumors ($n = 11$) in black or CARMIL1-AA tumors treated with vehicle ($n = 12$) or 5-FU ($n = 12$) in red.

subcutaneous tumor models[48], a result consistent with Fig. 7a–c. Most solid tumors, including breast cancers, contain necrotic areas, and evidence of tumor necrosis correlates with poor prognosis[13,49–51]. Our results with CARMIL1-AA 4T1 tumors provide compelling evidence that necrocytosis can make a significant contribution to tumor anabolism (Fig. 7 and Supplementary Fig. 7) and help to explain paradoxical observations that tumor cell death drives tumor growth.

Prior to this report, macropinocytosis had only been documented to provide amino acids[3–5,8–10,46,52]. While necrocytosis can maintain lipid droplet content in amino acid-deprived prostate cancer cells[3], whether this occurred because protein scavenging decreased the need for lipid droplet catabolism or because necrotic cell debris supplied lipids was unclear. Click chemistry-based flux analysis now confirms that many macromolecules, not just amino acids, can be recovered via necrocytosis. Extracellular proteins are often glycosylated; many macropinocytosed proteins would carry sugars that could be recycled. Carbohydrate scavenging (Supplementary Fig. 3c, d) could spare glucose that would otherwise be

required for glycosylation and stimulate pro-growth signal transduction and transcription. For example, GlcNAc synthesis is important for the expression of growth factor receptors that are critical for breast cancer cell proliferation and survival[53,54]. Fatty acids were also scavenged from necrotic cell debris via macropinocytosis (Fig. 3c, d). Fatty acids can be oxidized, used for membrane synthesis, or support signaling in growing cancer cells. Fatty acid synthesis is particularly important in breast and prostate cancer cells, and fatty acids synthase inhibitors are in clinical trials[30,31]. The effectiveness of the FASNi GSK2194069 against breast and prostate cancer cells was significantly compromised if macropinocytic cells had access to necrotic cell debris (Fig. 3e–i); if fatty acids can be scavenged from the tumor microenvironment, cells will no longer be dependent on FASN. In sum, necrocytosis is likely to decrease the effectiveness of a wide range of drugs targeting tumor anabolism by providing the end products of biosynthesis. Pairing these metabolic therapies with macropinocytosis inhibitors may increase the depth of the response and limit the development of resistance.

Therapeutic resistance is a major barrier to effective cancer therapy, particularly in patients with aggressive tumors (e.g., pancreas, prostate, and triple-negative breast cancer) where mutations that drive macropinocytosis are common. In keeping with the results presented here (Figs. 5 and 7), patients bearing tumors with *PIK3CA* or *KRAS* mutations or with decreased *PTEN* activity are more likely to be resistant to chemotherapy[55–58]. There is also a strong link between tumor necrosis and therapeutic resistance across tumor classes. Necrosis would provide high-quality macropinocytic fuel, reducing dependence on nucleotide biosynthesis pathways that are a known therapeutic liability[14,15,59,60]. It is particularly striking that the nucleotide synthesis inhibitors 5-FU and gemcitabine were ineffective if cells were able to perform necrocytosis (Figs. 5a–d, i, and j, 6e and Supplementary Fig. 6d) translating into significant therapeutic resistance in macropinocytic tumors in vivo (Fig. 7e, f and Supplementary Fig. 7e). This result is reminiscent of recent reports that deoxycytidine release from macrophages also limits the effectiveness of gemcitabine[61]. The effectiveness of genotoxic therapies such as doxorubicin and γ-irradiation that create dependence on de novo nucleotide synthesis pathways[15,62] was also compromised by necrocytosis (Fig. 5e–g). Genotoxic therapies and radiation are standard-of-care treatments for many cancer classes that are likely to be macropinocytic. Moreover, therapy may induce macropinocytosis in some tumors (Fig. 5h). Glioblastomas, a cancer class with a dismal long-term survival rate even with therapy, often have *PTEN* or *PIK3CA* mutations, AMPK activation, and large areas of necrosis at diagnosis[63,64]. Both radiation and temozolomide, an alkylating agent, are first line treatments; necrocytosis may play an important role in therapeutic resistance in glioblastoma patients. In summary, when used in combination with radiation and standard-of-care chemotherapy, macropinocytosis inhibitors have the potential to produce significant gains in survival in patients with lethal, aggressive cancers.

The contribution that macropinocytosis makes to cancer cell anabolism and therapeutic resistance has likely gone unrecognized in part due to the conditions under which in vitro experiments are generally performed. Standard tissue culture media are largely bereft of macropinocytic fuel, containing only limited amounts of albumin (10% serum provides ~0.3% albumin[3]). In contrast, the tumor microenvironment is rich in macromolecules and debris that are ripe for scavenging (Supplementary Fig. 7b). Indeed, macropinocytosis may provide one explanation why discrepant results are obtained when metabolic inhibitors are used in vitro and in vivo[65]. An additional translational implication of this study is that the clinical benefits of autophagy inhibitors that block lysosomal degradation (e.g. chloroquine derivatives[66,67]) may be derived as much from blocking macropinocytic flux as from autophagy inhibition. If so, the biomarkers selected to identify sensitive patients and confirm therapeutic efficacy would need to be reconsidered. Given the accumulating evidence that many tumor classes are macropinocytic and the clear anabolic benefits of scavenging, it is very likely that macropinocytosis makes as large a contribution to therapeutic resistance as does autophagy. Although additional studies will be required to fully assess the selectivity of the CARMIL1-AA mutant for macropinocytosis among other actin dependent processes, CARMIL1-AA is currently the only chemical or genetic approach to blocking macropinocytosis that preserves normal proliferation in non-macropinocytic cells. By offering a more selective means to disable macropinocytosis in tumor cells than EIPA (Fig. 6), the CARMIL1-AA mutant will facilitate future studies evaluating the role of macropinocytosis in tumor growth and progression.

## Methods

**Cell lines and cell culture.** All cultured cells were maintained at 37 °C in 5% $CO_2$. All media were supplemented with 10% standard fetal bovine serum and antibiotics unless otherwise stated. MDA-MB-231, MDA-MB-468, MCF-7, T-47D, BT-549, Hs578T, HCC1569, hTERT-HME1 PANC-1, BxPC3 and LNCaP, cells were obtained from the ATCC. The 4T1 cell line was provided by Jennifer Prescher (UC Irvine) and MCF10A PIK3CA knock-in cells[20] were supplied by Ben Ho Park (Johns Hopkins School of Medicine); both 4T1 and MCF10A cells were originally purchased from the ATCC. 22Rv1 cells were provided by Ionis Pharmaceuticals (Carlsbad, CA). hTERT-HME1 were cultured without serum in MEGM base medium supplemented additives provided in the MEGM™ BulletKit™ medium. MCF10A cells were cultured in DMEM/F12 Ham's Mixture without phenol red and supplemented with 5% horse serum, EGF (20 ng/ml), insulin (10 mg/ml), hydrocortisone (0.5 mg/ml), cholera toxin (100 ng/ml), 1% penicillin and streptomycin. MDA-MB-231, MDA-MB-468, and PANC-1 cells were cultured in DMEM with L-glutamine, 4.5 g/L glucose and without sodium pyruvate and supplemented with 1% sodium pyruvate. MCF-7 cells were cultured in RPMI supplemented with 1% L-glutamine. LNCaP, HCC1569, 22Rv1, and BxPC3 cells were cultured in RPMI-ATCC modified medium. T-47D and BT-549 cells were cultured in RPMI-ATCC modified medium with 0.2 or 0.023 IU/ml bovine insulin, respectively. 4T1 were cultured in DMEM with L-glutamine, 4.5 g/L glucose and without sodium pyruvate. Hs578T were cultured in the same medium as 4T1 but supplemented with 15% FBS and 0.01 μg/ml insulin. FL5.12 cells were maintained in RPMI 1640 medium supplemented with 10 mM HEPES, 55 μM β-mercaptoethanol, antibiotics, 2 mM L-glutamine, and 500 pg/ml murine rIL-3. $PIK3CA^{H1047R}$ and $PIK3CA^{E545K}$ MEFs were produced by transduction of p53$^{-/-}$ MEFs (generated in house) using pBabe-puro-HA-*PIK3CA*$^{H1047R}$ or -*PIK3CA*$^{E545K}$ (a gift from Jean Zhao, Addgene plasmids #12524 and #12525)[68]. All cells were passaged for ≤3 weeks at which point low-passage vials were thawed. *Mycoplasma* testing was performed using the VENOR GeM PCR kit every 4–6 months for all cell lines. Before use in these experiments, 4T1 cells were cured of *Mycoplasma* by culturing in ciprofloxacin for 8 wks; cure was confirmed by two serial PCR tests. All CARMIL1 reagents were obtained from John Cooper (Washington University). pFLRu-CARMIL1 shRNA-#1968 was used to knockdown endogenous CARMIL1 as described in ref. [69] (targeting sequence ATGCCATTGTTCATCTGGAT). A non-targeting shRNA (Addgene plasmid #1864, hairpin sequence CCTAAGGTTAAGTCGCCCTCGC TCGAGCGAGGGCGACTTAACCTTAGG) was used as a negative control as described in ref. [70]. Knockdown of endogenous CARMIL1 in 4T1 was validated by qRT-PCR (Forward primer: 5′-GAGCTGAGGTCAGGAGGAGC-3′ and Reverse primer: 5′-TTTTGCCCAATGCCAGGTGC-3′). shRNA-resistant cDNAs expressing wild type CARMIL1 (pFLRu-CARMIL1-WT shRNA-#1969) and CARMIL1 KR987/989AA (pFLRu-CARMIL1-AA shRNA-#1970) were introduced into 4T1 CARMIL1 knockdown cells, and clones selected. shRNA-resistant human CARMIL1 was cloned using Forward primer: 5′- GCCGAATTCAATGACCGAGGA GAGCTCTGACGTTC-3′ and Reverse primer: 5′- GCCGGATCCTTACACAA AAATAAACTCTTTTTC-3′[69]. Four to eight independent clones were screened in dextran uptake assays with similar results. Experiments conducted with two representative clones are shown.

**Microscopy.** Fluorescence microscopy was performed on either a Yokogawa spinning disk confocal microscope using a ×100 oil objective (dextran and CFSE-labeled FL5.12 uptake assays) or ×40 water objective with ×1.5 magnification (click chemistry and in vivo Ficoll uptake). Open source software micro-manager (version 2.0) was used to acquire images obtained with the spinning disk confocal. A Nikon Eclipse-Ti2000 inverted microscope was used for bright field imaging. All live confocal imaging (dextran or CFSE-labeled FL5.12 uptake assays) was performed at 37 °C. Z-stacks were collected with step size of 0.5 microns and 8–10 stacks were obtained from 20 to 40 independent fields. 5-FU-stimulated dextran uptake in Fig. 5 and all dextran uptake assays in Fig. 6 were performed in fixed cells and evaluated on a Zeiss LSM 780 confocal microscope using Zeiss Zen 2.3 image acquisition software. Microscope acquisition settings were held constant within each experiment and determined for each experiment using positive and negative controls. H&E sections were imaged using a Nikon Ti2-F inverted epi-fluorescence microscope equipped with a DS-Fi3 color camera (whole tumor sections) and a AmScope compound microscope with a Aptina MT9P031 color camera (high magnification).

**Dextran uptake assays.** Cells were seeded into 8-chamber slides (Cellvis, cat# C8-1.5H-N) 12–16 h before uptake assays. Cells were subjected to nutrient stress for 16 h prior to dextran uptake assays. MEM or RPMI containing 1% of the normal amount of amino acids and/or glucose (1% AA or 1% AA/gluc medium) was produced by preparing DMEM or RPMI lacking amino acids and/or glucose from chemical components and mixing it 99:1 with complete medium. Cells were incubated with 70 kD Oregon Green fluorescent dextran (1 mg/ml), 70 kD Texas Red fluorescent dextran (1 mg/ml), Lysotracker Red (500 nM) and Hoechst 33342 (1 μg/ml) for 30 min, washed three times with PBS, and fresh culture medium added. Drug pre-treatment and concentrations were as follows: EIPA, 50 μM, 1.5 h pre-treatment; PMA, 250 nM, co-addition with dextran; A769662, 50 μM, 1.5 h pre-treatment. Dextran uptake assays were performed 2 h after irradiation (2 Gy).

**Generation and uptake of necrotic and apoptotic cells**. Apoptotic FL5.12 cells were generated by withdrawing IL-3 from cells maintained at a density of 1 million/ml for 24 h. Necrotic cell debris was collected after 72 h of IL-3 withdrawal from cells maintained at 10 million/ml. Where indicated, FL5.12 cells were labeled with 5 µM CFSE at 500,000 cells/ml in PBS containing 1% FBS and 500 pg/ml IL-3 for 30 min. One million necrotic or apoptotic CFSE-labeled cell equivalents were spun down in a microfuge at 9000 × g at 4 °C for 10 min, the supernatant discarded, and the pelleted debris added to nutrient-deprived (1% AA/gluc for 16 h) macropinocytic breast cancer cells for 1 h. Where indicated, breast cancer cells were pre-treated with EIPA (50 µM) for 1.5 h. Cells were imaged live in fresh culture medium after 4–6 washes with PBS to remove any remnants of apoptotic or necrotic cells.

**Proliferation assays**. Cells at 60% confluence (12–16 h after seeding in a 24 well plate) were washed twice with PBS then incubated in 1% AA or 1% AA/gluc medium and supplemented with 10 million necrotic cell equivalents (0.2% protein), 10 million apoptotic cell equivalents (1% protein), or 5% fatty acid free BSA. The degree of nutrient stress was selected based on nutrient titration experiments demonstrating a 50% reduction in cell viability at 48 h. For proliferation assays in Figs. 4 and 6, cells were treated when 30% confluent (12–16 h after seeding in a 24 well plate) with: FASNi GSK2194069 (20 µM), 5-FU (1–30 µM), gemcitabine (20 µM), docetaxel (5 µM), or doxorubicin (1 µM) or subjected to γ-irradiation (5 Gy). Treated cells were provided with 10 million necrotic cell equivalents (0.2% protein) or left unsupplemented. For proliferation assays in Supplementary Figs. 4 and 5, GSK2194069-treated cells were supplemented with linoleic acid and oleic acid (1 mg/ml) conjugated to 1% albumin. 5-FU treated cells were supplemented with deoxythymidine monophosphate (1 µM). In Figs. 1c, g, 3e, h, 5a, c, 5e, g, and Supplementary Figs. 3f, g and 5d, cell proliferation was determined by flow cytometry by recording the number of cells that excluded vital dye (DAPI (1 mg/mL) or PI (1 mg/mL)) over a fixed collection interval (30 s). The gating strategy used to obtain live cell counts is illustrated in Supplementary Fig. 8. Representative bright field images were obtained before cells were processed for flow cytometry. In Figs. 5i, 6a, d, e, and Supplementary Figs. 4b, c, e, 5a, b, 6a, c, d, and 7c, d, proliferation was measured using crystal violet assays. Cells were fixed in 4% PFA for 15 min, stained with 0.05% crystal violet solution in PBS for 30 min, washed three times with 1X PBS to remove excess dye, and allowed to dry for 16–24 h. Crystal violet was recovered from cells using 100% methanol and absorbance at 595 nm measured using Gen5 version 5 microplate software reader.

**Flux analysis using click chemistry**. FL5.12 cells were labeled with 100 µM L-homopropargylglycine hydrochloride (HPG), 500 µM alkynyl palmitic acid (alk-PA), 100 µM N-(4-pentynoyl)-glucosamine tetraacylated (Ac₄GlcNAlk), or 1 mM EdU for 24 h at a density of 10 million/ml. In FL5.12 cells, HPG labeling was performed in methionine-free medium and alkynyl palmitic acid labeling was conducted in medium containing 10% charcoal stripped serum. Necrosis was induced in labeled cells as described above. Cancer cells (70% confluence in 8 chamber slides) were maintained in 1% AA medium for 16 h before 1 million cell equivalents of alkyne-labeled necrotic debris was added for 24–48 h. Cells were washed 3–5 times with PBS, fixed in 4% paraformaldehyde (10 min, RT), washed twice with PBS, permeablized and blocked (10% FBS and 0.3% Triton-X100 in PBS for 30 min rocking at RT), and washed twice with PBS. The click reaction (copper-catalyzed cycloaddition) was performed using 100 µM Tris [(1-benzyl-1H-1,2,3-triazol-4-yl)methyl] amine TBTA, 1 mM sodium ascorbate (made fresh each time), 100 µM CuSO₄ and 0.5 mM biotin-azide in 100 µl blocking solution for 1 h at 30 °C. After three washes with PBS, cells were incubated for 1 h with Alexa488-streptavidin (1:1,000 in blocking solution), stained with DAPI (1 mg/ml in PBS) for 10 min, washed, and imaged in PBS. Drug concentrations and pre-treatments were as follows: cycloheximide, 50 µg/mL (4 h); hydroxyurea, 10 mM, (4 h); or EIPA, 50 µM (1.5 h).

**In vivo experiments**. All experiments in mice were performed in accordance with protocols approved by the Institutional Animal Care and Use Committee of University of California, Irvine. To produce orthotopic tumors, 10⁴ 4T1 cells in 100 µl PBS were injected into the 4th mammary fat pad of 3–4-week-old female BALB/c mice. Twenty-three days after inoculation (tumor volume ≤ 1100 mm³), 70 kD FITC-Ficoll (1 mg dissolved in 1% Evan's Blue Dye in normal saline) was injected intra-tumorally 1 h after intraperitoneal (i.p.) injection of vehicle (DMSO) or 10 mg/kg EIPA. Mice were sacrificed 1 h after Ficoll injection, and tumors excised and frozen in OCT. Sections (5–8/tumor) were prepared by the Pathology Research Services Core Facility at UC Irvine, were fixed in 4% paraformaldehyde, stained with 1 mg/mL DAPI in PBS, washed twice with PBS, and mounted using Vectashield mounting medium. To produce orthotopic tumors from CARMIL1-WT and CARMIL1-AA 4T1 cells, 5 × 10⁴ cells were inoculated and evaluated as described above except that FITC-Ficoll was delivered intravenously. Tumor volume was calculated using caliper measurements and the formula, volume (mm³) = length [mm] × (width [mm])² × 0.52. Tumor volumes were measured every day once palpable. Once tumors reached 100 mm³, animals were randomly assigned to either the vehicle (12% DMSO in PBS) or 5-FU (30 mg/kg administered intraperitoneally every third day for a total of three treatments) group (n = 10–13). Mice were sacrificed once tumors reached 800 mm³ or if signs of distress were detected. To evaluate tumor necrosis, CARMIL1-WT

tumors from three mice were fixed in Tissue-Tek® Xpress® Molecular Fixative (Sakura) for 24 h after which tissues were paraffin embedded, sectioned (5 microns), and stained with hematoxylin/eosin (H&E). Evaluation of necrosis was performed by Experimental Tissue Resource core Director and board-certified pathologist, Dr. Robert A. Edwards (UC Irvine).

**Mouse husbandry**. Female BALB/c mice obtained from Jackson Laboratory (stock no. 000651) were maintained in autoclaved Envigo Teklab corncob (1/8th inch) under SPF conditions and allowed to acclimate to laboratory conditions for at least 72 h prior to use in experiments. Mice were housed in Techniplast individual ventilated cages (≤5 mice per cage) on ventilated racks at 21 ± 1 °C on a 12 h light/dark cycle. Animals were fed Envigo Teklad 2020X global soy protein-free extruded rodent diet. Mice were euthanized with CO₂ followed by cervical dislocation per AVMA guidelines.

**Image analysis**. All image processing and analysis was performed using Image J software (v2.0, NIH). Dextran and CFSE-necrotic or -apoptotic cell uptake was quantified on a per cell basis, to determine the percent of cell area composed of macropinosomes. Background subtraction was performed using a rolling ball radius of 50 pixels. To segment macropinosomes, auto-thresholding was performed until all macropinosomes were covered by the red thresholding signal. The same thresholding values were applied across all cells within the experiment. Where required, macropinosome segmentation was corrected manually to avoid over or under compensation. The binary image was analyzed using the "analyze particles" tool to retrieve the "area index". For tumor samples, Ficoll uptake analysis was performed on a per field rather than per cell basis. Maximum projections of z-stacks were used for analysis and are shown in the figures. Microscopy data shown in Figs. 1d, 2b, c, 3a, b, 4a, b and Supplementary Figs. 2a, 3a, b and 4a was not quantified, representative images from two independent experiments are shown.

**Statistical analysis and reproducibility**. All statistical analysis was performed using Graphpad Prism v7.0. In box plots, the center line is the median, the box is delimited by the 25th to 75th percentile, and whiskers represent minimum and maximum values. When two groups were compared, significance was determined using a two-tailed, unpaired t test. When >2 groups were compared, a one-way ANOVA was employed. Tukey's method or Dunnett's test was used to correct for multiple comparisons where required. For survival curves, a two-sided Mantel-Cox logrank test was applied. Mean ± SEM are shown unless otherwise indicated. *P ≤ 0.05; **P ≤ 0.01; ***P ≤ 0.001, ns, not significant, P > 0.05. For all comparisons, actual P-values are reported in the data source file that accompanies this manuscript.

**Reporting summary**. Further information on research design is available in the Nature Research Reporting Summary linked to this article.

## Data availability
Data supporting the findings of this study are available within the paper, its supplementary information files, and the source data file or available from the authors upon request. A reporting summary for this article is available as a supplementary information file.

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

## Acknowledgements

The authors thank Ben Ho Park (Johns Hopkins School of Medicine), Jenn Prescher (UC Irvine), and Ionis Pharmaceuticals for generously providing MCF10A cells, 4T1 cells or 22Rv1 cells, respectively, and John Cooper (Washington University) for providing CARMIL1-related reagents and advice. We thank David Fruman, Wenqi Wang, Brendan Finicle, Seong Kim, Tricia Nguyen, Archna Ravi, and Alison McCracken for valuable discussions and Alison McCracken for reviewing the manuscript prior to submission. We thank Elizabeth Selwan for assistance with γ-irradiation and caliper measurements and Robert Edwards, Wenqi Wang, Scott Atwood, and Gayoung Seo for assistance with evaluation and imaging of H&E stained tumors. This work was supported by grants from the CDMRP (W81XWH-15-1-0010), the University of California Cancer Research Coordinating Committee (CRR-17-426826), the Chao Family Comprehensive Cancer Center Anti-Cancer Challenge, and UCI Applied Innovation. The authors wish to acknowledge the support of the Chao Family Comprehensive Cancer Center Optical Biology Core and the Experimental Tissue Resource, shared resources supported by the National Cancer Institute of the National Institutes of Health under award number P30CA062203. The content of this manuscript is solely the responsibility of the authors and does not necessarily represent the official views of the National Institutes of Health.

## Author contributions

The study was conceived by V.J. and A.L.E. Experiments were designed by V.J. and A.L.E. Experiments were carried out by V.J. Data were analyzed and interpreted by V.J. and A.L.E. The manuscript was prepared by V.J. and A.L.E.

## Competing interests

A.L.E. is an inventor on patent application 15/760199 which includes molecules that, among other activities, inhibit macropinocytic flux. V.J. declares no competing interests.
