## [Peer Review File · Nature Communications]

Reviewers' comments:

Reviewer #1 (Remarks to the Author):

In their manuscript, Jayashankar and Edinger describe whether macropinocytosis of necrotic cellular debris (necrocytosis) can support cancer cell proliferation and viability in response to numerous metabolic stresses. The authors demonstrate that breast cancer cells can undergo macropinocytosis largely dependent upon RAS/PI3K pathway activation or upon nutrient starvation. They go on to show that cancer cell lines capable of macropinocytosis, but not those which are not, can survive blockade of a diverse array of cell essential biosynthetic pathways when provided with necrotic material. Using click-able molecules, the authors demonstrate transfer of these various molecules from necrotic cell to target cell via macropinocytosis. The implications of this work are clear: necrocytosis may play an outsized role in cancer cell survival of chemotherapy in patients. I find the manuscript and data presentation to be clear and the experiments well-controlled.

Minor Comments:

1) The authors use EIPA to block uptake of labeled metabolites in many settings. However, they only show that EIPA can block the increase in cell viability that necrotic cells provide in response to radiation. It would greatly strengthen the manuscript if the authors could provide data showing that the impact of EIPA on viability +/- necrotic cells in other contexts (e.g. amino acid or palmitate deprivation).

2) The authors state, "EIPA-sensitive 70 kD FITC-Ficoll uptake was observed in orthotopic, syngeneic 4T1 tumors in female BALB/c mice indicating that the level of AMPK activation in tumors is sufficient to trigger macropinocytosis."

The authors should consider that other triggers of macropinocytosis (besides AMPK) may be operating in vivo.

Reviewer #2 (Remarks to the Author):

In this study, the author use labeling of amino acids, sugars, fatty acids and nucleic acids with an alkyne to show that macropinocytic breast cancer cells can scavenge various biomass components from dead cell debris. The also show that the uptake of fatty acids from necrotic cells can allow to overcome targeting of fatty acid synthesis, and suggest that nucleotide scavenging from dead cells can contribute to resistance to anti-metabolite chemotherapy.

These data build on a growing literature showing that non-specific scavenging of material via macropinocytosis can be a feature of some cancer cells to obtain nutrients. It is nice in demonstrating formally that this can extend to nutrients other than amino acids, as has been postulated in other work. Exactly how this relates to real tumors is not explored, which is fine, although these limitations should be discussed. There are also some additional overstatements that should be corrected, particularly for publication in a top journal.

1. What is unclear is whether there is enough necrotic cell debris in real tumors to be a meaningful source of nutrients. I appreciate this is a challenging thing to determine, but could be better discussed.

2. The cells likely recover fatty acids from dead cells beyond palmitate, particularly when supplying this fatty acid alone without unsaturated fatty acids can be toxic. One benefit of this mechanism would be to obtain fatty acids in the right physiological ratios.

3. While not really the topic of this study, the fact that secondary "necrotic" bits of apoptotic

FL5.12 cells is used in the assay suggests it is probably another overstatement to claim that there is something different about necrotic versus apoptotic cell debris other than size.

4. I wonder if it is also an overstatement to suggest that it is "surprising" that macropinocytosis can play a role in supporting cells in nutrient deprived conditions. This is the topic of several other studies. Of note, the possibility has been suggested in other work that scavenged cell debris can supply material beyond amino acids.

5. As a minor point, in the abstract they refer to tumor proliferating in "amino acid deficient media". This is a bit of a strange statement.

Reviewer #3 (Remarks to the Author):

The manuscript by Jayashankar and Edinger investigates the role of macropinocytosis in breast cancer proliferation. The hypothesis is that cancer cells use macropinocytosis to scavenge nutrients from the tumor microenvironment, and that these nutrients can both support biosynthetic pathways and drive resistance to common chemotherapeutics that target metabolism. Although the role of macropinocytosis in acquiring amino acids is known, its role in scavenging lipids and sugars is less well appreciated. This study advances a new approach to studying macropinocytosis that utilizes click chemistry and is tangible for any lab to perform. In that sense, it reads more like a methods paper. In summary, I think this is an interesting topic and an important area, but that it is not a strong candidate for Nature Communications at least in its current form.

The manuscript would be significantly strengthened if there was more evidence to support these findings *in vivo*, particularly in necrotic tumor regions. I realize this may be experimentally difficult, but some attempt, and more commentary is warranted. It is important to confirm that this is not a cell-culture artifact, and that targeting macropinocytosis *in vivo* would impair tumor growth.

The conceptual advance of this study over previous studies in pancreatic and prostate cancer should be more clearly articulated.

In many of the assays, necrotic cell debris is added. Because necrotic cell debris is undefined, it seems hard to determine specificity. For example, in the assays in which FASN and thymidylate synthase is being inhibited, could only lipids/nucleotides be added back to achieve the same rescue? Or could lipids/nucleotides be removed from the cell debris so that it no longer rescues?

Is there a way to confirm that HPG labeling is specifically on newly synthesized endogenous proteins? The cycloheximide experiment (Fig 2) only indicates that new protein synthesis is required.

RESPONSE TO REVIEW

We thank the reviewers for their valuable feedback. The manuscript has been substantially improved by responding to their comments. These responses are summarized below:

Reviewer #1

1) It would greatly strengthen the manuscript if the authors could provide data showing that the impact of EIPA on viability +/- necrotic cells in additional contexts (e.g. amino acid or palmitate deprivation).

We have added figures showing that EIPA blocks the rescue driven by necrotic debris both in 1% AA/glucose and the presence of 5-FU (Figs. S3g and S5a). However, we have also added figures illustrating that EIPA has significant, macropinocytosis-independent effects (Fig. 6a and S6a). As a more specific strategy to address the underlying question – to what extent are the benefits of necrotic cells dependent on macropinocytosis – we now show that selective, genetic macropinocytosis inhibition using CARMIL1-AA completely eliminates the protection afforded by necrotic debris in low nutrients or in the presence of 5-FU (Fig. 6b-e and S6b-d). Unlike EIPA (Fig. 6a), CARMIL1-AA does not limit proliferation under conditions where cells are not macropinocytic (Fig. 6d, complete medium).

2) The authors state, “EIPA-sensitive 70 kD FITC-Ficoll uptake was observed in orthotopic, syngeneic 4T1 tumors in female BALB/c mice indicating that the level of AMPK activation in tumors is sufficient to trigger macropinocytosis.” The authors should consider that other triggers of macropinocytosis (besides AMPK) may be operating in vivo.

The text has been changed to “. . . indicating that AMPK activation or other signals are sufficient to trigger macropinosome formation in vivo (Fig. 1b)²³.”

Reviewer #2

1. What is unclear is whether there is enough necrotic cell debris in real tumors to be a meaningful source of nutrients. I appreciate this is a challenging thing to determine, but could be better discussed.

The new Figs. 6 and 7 and Supplementary Figs. 6 and 7 provide evidence that there is sufficient necrotic cell debris in tumors to drive cell growth. We also show that albumin, an alternative macropinocytic fuel, is not sufficient to drive proliferation in low nutrient medium (Fig. 1c, S3f, & S7c) or rescue cells from 5-FU (Fig. S7d). Necrotic cell debris, in contrast, supports proliferation under these conditions. The finding that macropinocytosis supports both 4T1 tumor growth and affords resistance to 5-FU in vivo (Fig. 7 and S7) therefore strongly suggests that there is enough necrotic cell debris available in the microenvironment to make a significant contribution to tumor anabolism. Evidence that 4T1 tumors are highly necrotic (Supplementary Fig. 7b) is consistent with published reports. As requested, these points are now more fully discussed in the text.

2. The cells likely recover fatty acids from dead cells beyond palmitate, particularly when supplying this fatty acid alone without unsaturated fatty acids can be toxic. One benefit of this mechanism would be to obtain fatty acids in the right physiological ratios.

We agree and thank the review for pointing this out. This is now stated in the text associated with Fig. 3.

3. While not really the topic of this study, the fact that secondary “necrotic” bits of apoptotic FL5.12 cells is used in the assay suggests it is probably an another overstatement to claim that there is something different about necrotic versus apoptotic cell debris other than size.

We agree with the reviewer – macropinocytosis is a non-specific uptake process and apoptotic cells are likely excluded from macropinosomes only due to their size. We have clarified this in the revised text.

4. I wonder if is also an overstatement to suggest that it is “surprising” that macropinocytosis can play a role in supporting cells in nutrient deprived conditions. This is the topic of several other studies. Of note, the possibility has been suggested in other work that scavenged cell debris can supply material beyond amino acids.

The term “surprising” is no longer used. Significant revisions to the abstract and text should make the conceptual advances in this manuscript more obvious. To our knowledge, it has never been suggested that

nucleotides could be scavenged by macropinocytosis, a finding we demonstrate has major ramifications for resistance to standard of care therapeutics.

5. As a minor point, in the abstract they refer to tumor proliferating in “amino acid deficient media”. This is a bit of a strange statement.

We thank the reviewer for pointing out this misstatement. We have corrected the abstract.

Reviewer #3

The manuscript would be significantly strengthened if there was more evidence to support these findings in vivo, particularly in necrotic tumor regions. I realize this may be experimentally difficult, but some attempt, and more commentary is warranted. It is important to confirm that this is not a cell-culture artifact, and that targeting macropinocytosis in vivo would impair tumor growth.

The effect of inhibiting macropinocytosis in tumors is now shown (Figs. 7 and S7); the field will clearly benefit from this validation of a genetic strategy to inhibit macropinocytosis in cancer cells. As described in the response to Reviewer 2, additional evidence that necrocytosis is relevant in vivo has been provided.

The conceptual advance of this study over previous studies in pancreatic and prostate cancer should be more clearly articulated.

Conceptual advances over published studies include:

- 1) breast cancer cells can support anabolism with macropinocytosis
- 2) unlike pancreas and prostate cancer cells, breast cancer cells may be contextually macropinocytic (macropinocytic only under stress)
- 3) macropinocytosis provides lipids, sugars, and nucleotides, not just amino acids
- 4) necrotic cell debris can support proliferation in macropinocytic cells under conditions where albumin does not; studies with albumin may underestimate the contribution of macropinocytosis to anabolism
- 5) macropinocytosis affords resistance to inhibitors of anabolic pathways currently targeted in clinical trials (e.g. FASNi)
- 6) EIPA significantly inhibits the proliferation of non-macropinocytic cancer cells confounding the interpretation of experiments using this compound to dissect the role of macropinocytosis in cancer cell proliferation or tumor growth
- 7) macropinocytosis can be more selectively inhibited in cancer cells by replacing endogenous CARMIL1 with CARMIL-AA
- 8) macropinocytosis affords resistance to standard-of-care cancer therapies including 5-FU, doxorubicin, gemcitabine, and γ -irradiation
- 9) Genotoxic therapies can trigger macropinocytosis in contextually macropinocytic cells

The abstract and text have been revised to better convey how this manuscript vertically advances the field.

In many of the assays, necrotic cell debris is added. Because necrotic cell debris is undefined, it seems hard to determine specificity. For example, in the assays in which FASN and thymidylate synthase is being inhibited, could only lipids/nucleotides be added back to achieve the same rescue? Or could lipids/nucleotides be removed from the cell debris so that it no longer rescues?

We thank the reviewer for this helpful suggestion. Figs. S4b-f and S5b,c now demonstrate that fatty acids or nucleotide precursors rescue from FASNi or 5-FU, respectively.

Is there a way to confirm that HPG labeling is specifically on newly synthesized endogenous proteins? The cycloheximide experiment (Fig 2) only indicates that new protein synthesis is required.

Fig. S2 has been added to increase confidence in HPG labeling studies. Our previous report (reference 3) used SILAC labeling to definitively demonstrate amino acid flux from dead cells into the proteome of

macropinocytic DU145 cells. These experiments are now repeated with HPG to validate the new assay. Disrupting HPG labeling with CHX is a rigorous method to demonstrate that new protein synthesis is required for macropinocytic cells to incorporate scavenged amino acids into their proteome. CHX blocked HPG incorporation in both MCF-7 and DU145 cells.

REVIEWERS' COMMENTS:

Reviewer #1 (Remarks to the Author):

The authors have addressed, in outstanding fashion, a challenging set of reviews. I anticipate that this work will be of principle importance to the tumor metabolism field. I have no further comments.

Reviewer #2 (Remarks to the Author):

The efforts to improve the study are noted, although they now rely heavily on the CARMIL1 alleles to argue for what is and is not macropinocytosis dependent in tumors. While they cite a paper for this claim, and confirm that these alleles do indeed perturb macropinocytosis, I doubt this is the only affect of the alleles on cell biology and some will take issue with this specifically dissecting macropinocytosis in vivo. It would be unfair to expect the authors to develop a macropinocytosis-specific tool when this remains a problem for the field, but not all readers are as thoughtful and I still encourage them to make the limitations of these tools more clear.

With this said, I still think it is a nice paper and should be published.

Reviewer #3 (Remarks to the Author):

The authors have addressed my concerns.

RESPONSE TO REVIEW

We thank the reviewers for participating in peer review and for their valuable feedback throughout the review process. Our responses to their final comments are summarized below:

Reviewer #1 (Remarks to the Author):

The authors have addressed, in outstanding fashion, a challenging set of reviews. I anticipate that this work will be of principle importance to the tumor metabolism field. I have no further comments.

We thank the reviewer for these positive comments on the revised manuscript.

Reviewer #2 (Remarks to the Author):

The efforts to improve the study are noted, although they now rely heavily on the CARMIL1 alleles to argue for what is and is not macropinocytosis dependent in tumors. While they cite a paper for this claim, and confirm that these alleles do indeed perturb macropinocytosis, I doubt this is the only affect of the alleles on cell biology and some will take issue with this specifically dissecting macropinocytosis in vivo. It would be unfair to expect the authors to develop a macropinocytosis-specific tool when this remains a problem for the field, but not all readers are as thoughtful and I still encourage them to make the limitations of these tools more clear.

With this said, I still think it is a nice paper and should be published.

We appreciate this thoughtful comment and agree that the CARMIL1-AA mutant is a useful tool, but not without limitations. However, as recognized by the reviewer, selectively inhibiting macropinocytosis remains a challenge for the field. CARMIL1-AA is completely unable to support macropinocytosis, but does not compromise proliferation in complete medium. No chemical inhibitor or alternative genetic strategy reported to date meets this standard. We have added a sentence to the discussion stating that additional work will be required to more fully assess the selectivity of CARMIL1-AA for macropinocytosis over other actin-dependent processes.

Reviewer #3 (Remarks to the Author):

The authors have addressed my concerns.

We thank the reviewer for evaluating the revised manuscript.